# Rotamer Density Estimator is an Unsupervised Learner of the Effect of Mutations on Protein-Protein Interaction

**Shitong Luo**[1*]**, Yufeng Su**[2*]**, Zuofan Wu**[1]**, Chenpeng Su**[1]**, Jian Peng**[1,2] **& Jianzhu Ma**[1,3]

[1] Helixon Research
[2] University of Illinois Urbana-Champaign
[3] Institute for AI Industry Research, Tsinghua University
`luost@helixon.com,luost26@gmail.com`
`jianpeng@illinois.edu,majianzhu@tsinghua.edu.cn`

## Abstract

Protein-protein interactions are crucial to many biological processes, and predicting the effect of amino acid mutations on binding is important for protein engineering. While data-driven approaches using deep learning have shown promise, the scarcity of annotated experimental data remains a major challenge. In this work, we propose a new approach that predicts mutational effects on binding using the change in conformational flexibility of the protein-protein interface. Our approach, named Rotamer Density Estimator (RDE), employs a flow-based generative model to estimate the probability distribution of protein side-chain conformations and uses entropy to measure flexibility. RDE is trained solely on protein structures and does not require the supervision of experimental values of changes in binding affinities. Furthermore, the unsupervised representations extracted by RDE can be used for downstream neural network predictions with even greater accuracy. Our method outperforms empirical energy functions and other machine learning-based approaches.

## 1 Introduction

Proteins rarely act alone and usually interact with other proteins to perform a diverse range of biological functions (Alberts & Miake-Lye, 1992; Kastritis & Bonvin, 2013). For example, antibodies, a type of immune system protein, recognize and bind to proteins on pathogens' surfaces, eliciting immune responses by interacting with the receptor protein of immune cells (Lu et al., 2018). Given the importance of protein-protein interactions in many biological processes, developing methods to modulate these interactions is critical. A common strategy to modulate protein-protein interactions is to mutate amino acids on the interface: some mutations enhance the strength of binding, while others weaken or even disrupt the interaction (Gram et al., 1992; Barderas et al., 2008). Biologists may choose to increase or decrease binding strength depending on their specific goals. For example, enhancing the effect of a neutralizing antibody against a virus usually requires increasing the binding strength between the antibody and the viral protein. However, the combinatorial space of amino acid mutations is large, so it is not always feasible or affordable to conduct wet-lab assays to test all viable mutations. Therefore, computational approaches are needed to guide the identification of desirable mutations by predicting their mutational effects on binding strength, typically measured by *the change in binding free energy* ($\Delta\Delta G$).

Traditional computational approaches are mainly based on biophysics and statistics (Schymkowitz et al., 2005; Park et al., 2016; Alford et al., 2017). Although these methods have dominated the field for years, they have several limitations. Biophysics-based methods face a trade-off between efficiency and accuracy since they rely on sampling from energy functions. Statistical methods are more efficient, but their capacity is limited by the descriptors considered in the model. Furthermore, both biophysics and statistics-based methods heavily rely on human knowledge, preventing it to

---

*Equal contribution.

benefit from the growing availability of protein structures. As a result, predicting the effects of mutations on protein-protein binding remains an open problem.

Recently, deep learning has shown significant promise in modeling proteins, making data-driven approaches more attractive than ever (Rives et al., 2019; Jumper et al., 2021). However, developing deep learning-based models to predict mutational effects on protein-protein binding is challenging due to the scarcity of experimental data. Only a few thousand protein mutations, annotated with changes in binding affinity, are publicly available (Geng et al., 2019b), making supervised learning challenging due to the potential for overfitting with insufficient training data. Another difficulty is the absence of the structure of mutated protein-protein complexes. Mutating amino acids on a protein complex leads to changes mainly in sidechain conformations (Najmanovich et al., 2000; Gaudreault et al., 2012), which contribute to the change in binding free energy. However, the exact conformational changes upon mutation are unknown.

In this work, we draw inspiration from the thermody-namic principle that protein-protein binding usually leads to entropy loss on the binding interface, which can be used to determine binding affinity (Brady & Sharp, 1997; Cole & Warwicker, 2002; Kastritis & Bonvin, 2013). When two proteins bind, the residues located at the interface tend to become less flexible (i.e. having lower entropy) due to the physical and geometric constraints imposed by the binding partner (Figure 1). A higher amount of entropy loss corresponds to a stronger binding affinity. Therefore, by comparing the entropy losses of wild-type and mutated protein complexes, we can estimate the effect of mutations on binding affinity. Please refer to Section B in the appendix for a detailed discussion.

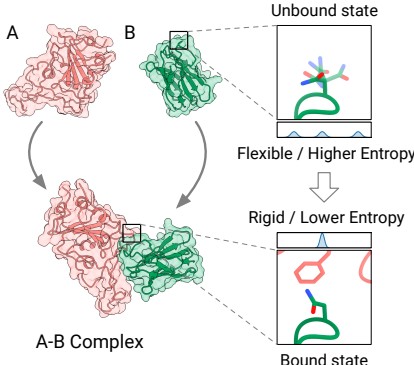

Figure 1: The conformational flexibility of the interface generally decreases upon binding.

Based on this principle, we introduce a novel approach to predict the impact of amino acid mutations on protein-protein interaction. The core of our method is Rotamer Density Estimator (RDE), a conditional generative model that estimates the density of amino acid sidechain conformations (rotamers). We use the entropy of the estimated density as a metric of conformational flexibility. By subtracting the entropy of the separated proteins from the entropy of the complex, we obtain an estimation of binding affinity. Finally, we can assess the effect of mutations by comparing the estimated binding affinities of wild-type and mutant protein complexes. In addition to directly comparing entropy, we also employ neural networks to predict $\Delta\Delta G$ from the representations learned by RDE.

Our method is an attempt to address the aforementioned challenges. Rotamer Density Estimator is trained solely on protein structures, not requiring other labels, making it an unsupervised learner of the mutation effect on protein-protein interaction. This feature mitigates the challenge posed by the scarcity of annotated mutation data. Moreover, our method does not require the mutated protein structure as input. Instead, it treats mutated structures as latent variables, which are approximated by RDE. Our method outperforms both empirical energy functions and machine learning models for predicting $\Delta\Delta G$. Additionally, as a generative model for rotamers, RDE accurately predicts sidechain conformations.

## 2 RELATED WORK

### 2.1 MUTATIONAL EFFECT PREDICTION FOR PROTEIN-PROTEIN INTERACTION

Traditional approaches to predicting the effect of mutation on protein binding can be roughly divided into two classes: biophysical and statistical methods. Biophysical methods utilize energy functions to model inter-atomic interactions. These methods sample conformations of the mutated protein complex and predict changes in binding free energy (Schymkowitz et al., 2005; Park et al., 2016; Alford et al., 2017; Steinbrecher et al., 2017). Statistical methods rely on feature engineering, which uses descriptors summarizing geometric, physical, evolutionary, and motif properties of proteins to predict mutational effects (Geng et al., 2019a; Zhang et al., 2020). Traditional methods face the

trade-off between speed and accuracy. Their performance depends heavily on human expertise, which limits their pace to improve with the fast-growing of available protein structures.

Recently, deep learning-based approaches have emerged. We group them into three categories: end-to-end models, pre-training-based models, and unsupervised models. End-to-end models directly predict the difference in binding free energy by taking both mutant and wild-type protein structures as input (Shan et al., 2022). Pre-training-based models attempt to address data scarcity by pre-training a feature extraction network (Liu et al., 2021; Yang et al., 2022; Zhang et al., 2022). However, most pre-training tasks are not designed to capture the foundation of protein-protein interactions. Unsupervised models adopt the mask-predict paradigm to protein 3D structures, partially masking amino acid types on a given protein backbone, and recovering the masked information using neural networks (Wang et al., 2018; Shroff et al., 2020; Jing et al., 2020; Yang et al., 2022; Hsu et al., 2022). These models can serve as unsupervised predictors of the mutational effects on binding, as the difference in the probability of amino acid types before and after mutation correlates mildly to the change in binding free energy.

## 2.2 MUTATIONAL EFFECT PREDICTION FOR SINGLE PROTEINS

The prediction of mutational effects for single proteins can be achieved using either structure-based or sequence-based (evolution-based) approaches. Structure-based methods can be categorized into biophysical, statistical, and deep learning-based methods, which aim to predict the thermal stability or fitness of the protein rather than the binding free energy between proteins (Schymkowitz et al., 2005; Park et al., 2016; Alford et al., 2017; Lei et al., 2023). Sequence-based methods rely on the mining of evolutionary history, done by performing statistics on multiple sequence alignments (MSAs) constructed from large-scale sequence databases (Hopf et al., 2017; Riesselman et al., 2018; Rao et al., 2021; Luo et al., 2021; Frazer et al., 2021), or leveraging protein language models (PLMs) (Meier et al., 2021; Notin et al., 2022).

However, it is important to note that sequence-based methods are not suitable for predicting mutational effects on general protein-protein interactions due to the lack of evolutionary information in many cases. Protein-protein interactions typically involve two or more chains, which may belong to different species or may not experience inter-chain co-evolution. As such, it is infeasible to predict mutational effects via mining sequence databases using existing powerful tools such as MSAs or PLMs. Thus, effective ways for predicting mutational effects on protein-protein interaction rely on structure-based approaches rather than sequences alone. We present a detailed discussion and supporting experimental results in Section C.1 of the appendix.

## 3 METHOD

### 3.1 OVERVIEW AND PRELIMINARIES

**Overview** Our method comprises three main components. The first is Rotamer Density Estimator (RDE), which is a conditional normalizing flow that models the probability density of sidechain conformations (rotamers) based on the amino acid type and backbone structures (Section 3.2). The second component is an algorithm that estimates the entropy of the distribution parameterized by the normalizing flow (Section 3.3). Lastly, we describe how we use the entropy of the protein-protein interfaces in both the mutated and wild-type states, both bound and unbound, to estimate the change in binding free energy ($\Delta\Delta G$). We also detail how we use neural networks to achieve more accurate predictions of $\Delta\Delta G$ using the unsupervised representations from RDE (Section 3.4).

**Definitions and Notations** A *protein-protein complex* is a multi-chain protein structure that can be divided into two groups. Each group contains at least one protein chain and each chain consists of multiple (amino acid) residues. For a protein complex containing $n$ residues, we number them from 1 to $n$. The two groups of the complex can be represented by two disjoint sets of indices $A, B \subset \{1 \ldots n\}$. A residue is characterized by its type, position, orientation, and sidechain conformation. We denote the type, position, and orientation of the $i$-th ($i \in \{1 \ldots n\}$) residue as $a_i \in \{1 \ldots 20\}$, $\boldsymbol{p}_i \in \mathbb{R}^3$, and $\boldsymbol{O}_i \in SO(3)$ respectively. The sidechain conformation of the residue is called *rotamer*. As the conformational degree of freedom of the sidechain is defined by rotatable bonds, a rotamer can be parameterized by torsional angles w.r.t. the rotatable bonds. The number of torsional angles varies between 0 to 4 depending on the residue type. For a residue with $d$ torsional angles, we denote the $k$-th ($k \in \{1 \ldots 4\}$) torsional angle by $\chi_i^{(k)} \in [0, 2\pi)$. Collectively, all the torsional angles are

denoted by a vector $\boldsymbol{\chi}_i = (\chi_i^{(k)})_{k=1}^d$. Using the language of geometry, an angle can be represented by a point on the unit circle $\mathbb{S}^1$. A vector consisting of $d$ angular values resides on the product of $d$ unit circle, known as the $d$-dimensional torus $\mathbb{T}^D = (\mathbb{S}^1)^D$.

In this work, our first goal is to model the conditional probability density of rotamers, given the type, position, orientation, and prior rotamer of itself and other residues: $p(\boldsymbol{\chi}_i \mid \{a_j, \boldsymbol{p}_j, \boldsymbol{O}_j, \tilde{\boldsymbol{\chi}}_j\}_{j=1}^n)$. The prior rotamers $\tilde{\boldsymbol{\chi}}_j$ are often inaccurate or unknown. For example, if we mutate some residues, the rotamers of the mutated residues are unknown, and the rotamers of residues nearby the mutated ones are inaccurate because they are affected by the mutation. The probability density is defined over the $d$-dimensional torus $\mathbb{T}^D = (\mathbb{S}^1)^D$, and we describe below the flow-based architecture to model the density.

## 3.2 ROTAMER DENSITY ESTIMATOR

Rotamer Density Estimator (RDE) is designed to estimate the conditional distribution of the rotamer of the $i$-th residue, given the information of itself and other residues: $p(\boldsymbol{\chi}_i \mid \{a_j, \boldsymbol{p}_j, \boldsymbol{O}_j, \tilde{\boldsymbol{\chi}}_j\}_{j=1}^n)$. In this section, we first introduce the encoder network that produces hidden representations for each residue, taking into account its environment $\{a_i, \boldsymbol{p}_i, \boldsymbol{O}_i, \tilde{\boldsymbol{\chi}}_i\}_{i=1}^n$. Next, we present a conditional normalizing flow defined on $\mathbb{S}^1$ for rotamers with only 1 torsional angle. Based on the $\mathbb{S}^1$ flow, we further extend the flows to $d$-dimensional torus $\mathbb{T}^D$ ($D > 1$) for rotamers with more than 1 torsional angle.

**Encoder Network**  The encoder network starts with two multi-layer perceptrons (MLPs) that generate embeddings for each individual single residue and each pair of residues respectively. The MLP for single residues encodes the residue type, backbone dihedral angles, and local atom coordinates into a vector $\boldsymbol{e}_i$ ($i = 1 \dots n$). The other MLP for residue pairs encodes the distance and the relative position between two residues. We denote a pair embedding vector as $\boldsymbol{z}_{ij}$ ($i, j = 1 \dots n$). To transform the single embeddings $\boldsymbol{e}_i$ and pair embeddings $\boldsymbol{z}_{ij}$ into hidden representations $\boldsymbol{h}_i$, we use a self-attention-based network that is invariant to rotation and translation (Jumper et al., 2021). The hidden representation $\boldsymbol{h}_i$ aims to capture both the information of the $i$-th residue itself and its structural environment. It serves as an encoding of the condition $\{a_j, \boldsymbol{p}_j, \boldsymbol{O}_j, \tilde{\boldsymbol{\chi}}_j\}_{j=1}^n$ for the probability density with respect to $\boldsymbol{\chi}_i$.

**Conditional Flow on $\mathbb{S}^1$**  To model the distribution of a rotamer with 1 torsional angle, we utilize conditional normalizing flows on $\mathbb{S}^1$ (Rezende et al., 2020). A normalizing flow is a bijective function, and to construct one on $\mathbb{S}^1$, we parameterize it by an angle $\theta \in [0, 2\pi]$, and define a bijective function on $[0, 2\pi]$ that is equivalent to a bijective over $\mathbb{S}^1$. A common method for constructing a bijective function is to ensure strict monotonicity, which we adopt by constructing a strictly monotonically increasing function on $[0, 2\pi]$, denoted by $f : [0, 2\pi] \to [0, 2\pi]$. Notably, due to the periodicity of angular values, 0 and $2\pi$ are the same point, so to preserve continuity at both ends, we must ensure that $f(0) = 0$, $f(2\pi) = 2\pi$, and $f'(0) = f'(2\pi)$. To achieve this, we use the rational quadratic spline flow (Durkan et al., 2019; Rezende et al., 2020), a piece-wise function that contains $K$ pieces delimited by $K + 1$ knots. Each piece takes the form:

$$f_k(x|x_{k,k+1}, y_{k,k+1}, \delta_{k,k+1}) = y_k + \frac{(y_{k+1} - y_k)\left[s_k \xi_k^2(x) + \delta_k(1 - \xi_k(x))\xi_k(x)\right]}{s_k + \left[\delta_{k+1} + \delta_k - 2s_k\right](1 - \xi_k(x))\xi_k(x)}, \tag{1}$$

$$\text{where } s_k = \frac{y_{k+1} - y_k}{x_{k+1} - x_k}, \text{ and } \xi(x) = \frac{x - x_k}{x_{k+1} - x_k} \quad (x \in [x_k, x_{k+1}]), \tag{2}$$

where the spline is parameterized by the coordinates and derivatives of the $K + 1$ knots denoted by $x_k$, $y_k$, and $\delta_k$ ($k = 1 \dots K + 1$). To fulfill the requirement of monotonicity, continuity, and periodicity, we impose the following constraints on the knots: i. $0 = x_1 < x_2 < \dots < x_{K+1} = 2\pi$, ii. $0 = y_1 < y_2 < \dots < y_{K+1} = 2\pi$, iii. $\delta_k > 0(k = 1 \dots K + 1)$, and iv. $\delta_1 = \delta_{K+1}$. These parameters are produced by transforming the hidden representation $\boldsymbol{h}_i$ of residues using neural networks, so the probability distribution defined by the bijective is conditional on the residue and its environment. Therefore, we may also denote the spline by $f(x|\boldsymbol{h}_i)$.

We choose the uniform distribution on $[0, 2\pi]$ as the base distribution, represented by $p_z(z) = \frac{1}{2\pi}$ ($z \in [0, 2\pi]$). The mapping from the target distribution to the base distribution is denoted by $f$. According to the change-of-variable formula, the target rotamer density of the $i$-th residue is given by:

$$\log p(x|\boldsymbol{h}_i) = \log p_z(f(x)) + \log |f'(x|\boldsymbol{h}_i)| = -\log 2\pi + \log |f'(x|\boldsymbol{h}_i)|. \tag{3}$$

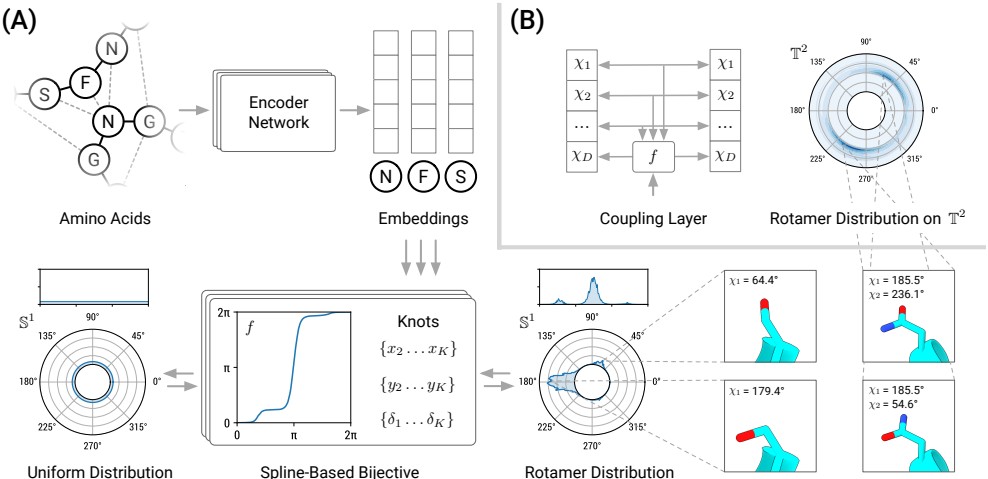

Figure 2: **(A)** The overall architecture of Rotamer Density Estimator (RDE) for estimating distributions of rotamers with one torsional angle. **(B)** Invertible coupling layers alternating between different dimensions enable modeling distributions of rotamers with multiple torsional angles.

The derivative $f'$ can be computed analytically according to Eq.1. See Figure 2A for an illustration of the model.

In practice, we stack multiple bijectives to enable more complex transformation. The derivative of the composite can be computed efficiently using the chain rule. At inference time, we can efficiently compute the inverse mapping $f^{-1}(y)$(Rezende et al., 2020): To find the solution of $f^{-1}(y)$, the first step is to locate the unique bin that contains $y$. Assuming $y$ belongs to the $k$-th bin, finding its corresponding $x$ amounts to finding the root of the quadratic equation $f_k(x|x_{k,k+1}, y_{k,k+1}, \delta_{k,k+1}) = y$ in the interval $[x_k, x_{k+1}]$, for which a closed-form solution exists.

**Conditional Flow on $\mathbb{T}^D$** Rotamers with $D$ torsional angles can be viewed as points on the $D$-dimensional torus which is the product of $D$ circles $\mathbb{S}^1$, i.e. $\mathbb{T}^D = \mathbb{S}^1 \times \cdots \times \mathbb{S}^1$. To model the distribution on $\mathbb{T}^D$, we adopt the coupling layer technique to model the joint distribution (Dinh et al., 2016). Each coupling layer updates one dimension using the bijective for $\mathbb{S}^1$, keeping the other $D - 1$ dimensions fixed, and uses the $D - 1$ dimensions along with the hidden representation of the residue as the conditioner to parameterize the bijective (Figure 2B):

$$g_d(\boldsymbol{x}|\boldsymbol{h}_i)[j] = \begin{cases} f(x_j|\boldsymbol{x}_{\setminus j}, \boldsymbol{h}_i), & j = d \mod D \\ x_j, & j \neq d \mod D \end{cases}, \tag{4}$$

where $d$ is the dimension to update. The coupling layer $g_d$ preserves invertibility and has closed-form inversion. The determinant of its Jacobian is equal to the derivative of $f(x_j|\boldsymbol{x}_{\setminus j}, \boldsymbol{h}_i)$. We stack multiple spline-based coupling layers to ensure that all the dimensions are updated at least once, resulting in a stack of $L$ coupling layers: $g = g_1 \circ g_2 \circ \cdots \circ g_L$ ($L \geq D$).

We choose the uniform distribution as the base distribution, denoted by $p_{\boldsymbol{z}}(\boldsymbol{z}) = (\frac{1}{2\pi})^D$ and let $g$ map the target distribution to the base distribution. The log-likelihood is computed using the change-of-variable rule:

$$\begin{aligned} \log p(\boldsymbol{x}|\boldsymbol{h}_i) &= \log p_{\boldsymbol{z}}(g(\boldsymbol{x})) + \log|\det \nabla_{\boldsymbol{x}} g(\boldsymbol{x})| \\ &= -D\log 2\pi + \sum_{l=1}^{L} \log\left|f_l'(x_{d(l)}|\boldsymbol{x}_{\setminus d(l)}, \boldsymbol{h}_i)\right|. \end{aligned} \tag{5}$$

**Training Objective** To train RDE, we minimize the negative log-likelihood of native (ground truth) rotamers:

$$\min \mathbb{E}_{(\boldsymbol{\chi}_i, \mathcal{S}) \sim p_{\text{data}}} \left[-\log p\left(\boldsymbol{\chi}_i|\boldsymbol{h}_i(\mathcal{S})\right)\right], \tag{6}$$

where $\mathcal{S} = \{a_j, \boldsymbol{p}_j, \boldsymbol{O}_j, \tilde{\boldsymbol{\chi}}_j\}_{j=1}^N$ is a protein structure sampled from the dataset. Since the model is designed for mutation analysis, the prior rotamers should emulate the rotamers after some residues are mutated. Thus, we mask the rotamers of a random portion of residues and add noise to the rotamers of the neighbors of the masked ones. The negative log-likelihood training objective function is evaluated on the masked residues and a random part of the perturbed residues.

### 3.3 ROTAMER ENTROPY ESTIMATION

RDE models the distribution of possible rotamers of residue, making the entropy of the rotamer distribution a natural measure of conformational flexibility (Brady & Sharp, 1997):

$$H(\mathcal{S}, i) = \mathbb{E}_{\boldsymbol{\chi} \sim p}\left[-\log p(\boldsymbol{\chi}|\boldsymbol{h}_i(\mathcal{S}))\right]. \tag{7}$$

To estimate the entropy, we use a stochastic method: First, we sample a set of rotamers from the distribution using the inverted flows (Eq.1, 4). Then, we compute the negative log probability of the samples and take their average as an estimate of the entropy. Computing these steps is efficient thanks to the ability to compute the exact likelihood of normalizing flows.

For residues without rotatable sidechains (alanine, glycine, and proline), we consider their conformational entropy to be constant. Nonetheless, we can still estimate the entropy of neighboring residues to evaluate their impact on conformational flexibility.

### 3.4 MUTATIONAL EFFECT ($\Delta\Delta G$) PREDICTION

Let us consider a protein-protein complex $\mathcal{W}_{LR} = \{a_i, \boldsymbol{p}_i, \boldsymbol{O}_i, \tilde{\boldsymbol{\chi}}_i\}$ consisting of $N$ residues, where $M$ of them belong to the first group $L = \{1, \ldots, M\}$, and the remaining $N - M$ belong to the second group $R = \{M + 1, \ldots, N\}$. We refer to group $L$ as the *ligand group* since it contains mutations, and group $R$ as the *receptor group*. In the unbound state, we represent the two separated structures by $\mathcal{W}_L$ and $\mathcal{W}_R$ respectively. If we mutate $m$ residues in group $L$, numbered by $\{1, \ldots, m\} \subset L$, the mutated structure in the bound state is denoted by $\mathcal{M}_{LR} = \{\tilde{a}_i, \boldsymbol{p}_i, \boldsymbol{O}_i, \varnothing\}_{i=1}^{m} \cup \{a_i, \boldsymbol{p}_i, \boldsymbol{O}_i, \tilde{\boldsymbol{\chi}}_i\}_{\text{unmutated}}$, where $\tilde{a}_i$ represents the mutated residue type and $\varnothing$ indicates that the rotamers of the mutated residues are unknown. The two groups in the unbound state are denoted by $\mathcal{M}_L$ and $\mathcal{M}_R$ respectively. Note that $\mathcal{M}_R = \mathcal{W}_R$ if no mutations are in the receptor group.

**Linear Predictor** The *entropy loss upon binding* of a protein complex $\mathcal{S}$ ($\mathcal{S} = \mathcal{W}, \mathcal{M}$) is defined as the difference of entropy between the bound and the unbound states (Kastritis & Bonvin, 2013). We approximate the entropy loss $\Delta H_{\mathcal{S}}$ using a linear model and our estimated entropy:

$$\Delta H_{\mathcal{S}} = \left[ w_{\mathcal{S}L}^{\text{bound}} \underbrace{\sum_{i \in L} (H(\mathcal{S}_{LR}, i) + E(a_i))}_{H_{\mathcal{S}L}^{\text{bound}} \text{ bound ligand}} + w_{\mathcal{S}R}^{\text{bound}} \underbrace{\sum_{i \in R} (H(\mathcal{S}_{LR}, i) + E(a_i))}_{H_{\mathcal{S}R}^{\text{bound}} \text{ bound receptor}} \right] -$$

$$\left[ w_{\mathcal{S}L}^{\text{unbnd}} \underbrace{\sum_{i \in L} (H(\mathcal{S}_L, i) + E(a_i))}_{H_{\mathcal{S}L}^{\text{unbnd}} \text{ unbound ligand}} + w_{\mathcal{S}R}^{\text{unbnd}} \underbrace{\sum_{i \in R} (H(\mathcal{S}_R, i) + E(a_i))}_{H_{\mathcal{S}R}^{\text{unbnd}} \text{ unbound receptor}} \right], (\mathcal{S} = \mathcal{W}, \mathcal{M}). \tag{8}$$

where $w_{\mathcal{S}J} > 0$ ($\mathcal{S} = \mathcal{W}, \mathcal{M}; J = L, R$) is the coefficient that controls the contribution of entropy to the binding free energy, and $E(\cdot)$ is the entropy bias for 20 different amino acid types. The thermodynamic definition states that the core component in binding $\Delta\Delta G$ is the difference in entropy loss between the wild-type and mutant structure. Therefore, we estimate $\Delta\Delta G$ by:

$$\begin{aligned}
\Delta\Delta G_{\text{pred}} &= \Delta H_{\mathcal{M}} - \Delta H_{\mathcal{W}} + b \\
&= (w_{\mathcal{M}L}^{\text{bound}} H_{\mathcal{M}L}^{\text{bound}} + w_{\mathcal{M}R}^{\text{bound}} H_{\mathcal{M}R}^{\text{bound}}) - (w_{\mathcal{M}L}^{\text{unbnd}} H_{\mathcal{M}L}^{\text{unbnd}} + w_{\mathcal{M}R}^{\text{unbnd}} H_{\mathcal{M}R}^{\text{unbnd}}) - \\
&\quad (w_{\mathcal{W}L}^{\text{bound}} H_{\mathcal{W}L}^{\text{bound}} + w_{\mathcal{W}R}^{\text{bound}} H_{\mathcal{W}R}^{\text{bound}}) + (w_{\mathcal{W}L}^{\text{unbnd}} H_{\mathcal{W}L}^{\text{unbnd}} + w_{\mathcal{W}R}^{\text{unbnd}} H_{\mathcal{W}R}^{\text{unbnd}}) + b.
\end{aligned} \tag{9}$$

Note that as we assume there are no mutations in the receptor group ($\mathcal{M}_R = \mathcal{W}_R$), $H_{\mathcal{M}R}^{\text{unbnd}}$ and $H_{\mathcal{W}R}^{\text{unbnd}}$ cancel out each other and do not contribute to $\Delta\Delta G$.

To calibrate the parameters in Eq.9, we use the block coordinate descent method, which alternates between optimizing the coefficients ($w_{\mathcal{S}J}$, $b$) and the entropy biases ($E$) using the mean squared error (MSE) loss.

**Neural Network Predictor** Each residue's hidden representation $\boldsymbol{h}_i$ used to parameterize the normalizing flows contains sufficient information about the rotamer distribution. To extract binding information from these representations in a more flexible way, we employed neural networks. Specifically, we utilized a network that shares the same architecture as the encoder to transform the representation $\boldsymbol{h}_i$ and applied max-pooling to obtain a global structure representation. We then subtracted

the representation of the wild-type structure from the mutant representation and fed it into another MLP to predict $\Delta\Delta G$. To enforce anti-symmetry, we swapped the wild-type and mutant to predict $-\Delta\Delta G$, and computed $(\Delta\Delta G - (-\Delta\Delta G))/2$ as the final prediction. The network was trained using the MSE loss. During training, *we freeze the weights of RDE and do not back-propagate gradients through $h_i$* to fully exploit the unsupervised representations learned by RDE.

### 3.5 MODEL TRAINING

The dataset for training RDE is derived from PDB-REDO (Joosten et al., 2014), which is a database containing refined X-ray structures in PDB. The protein chains are clustered based on 50% sequence identity, leading to 38,413 chain clusters, which are randomly divided into the training, validation, and test sets by 95%/0.5%/4.5% respectively. During training, the data loader randomly selects a cluster and then randomly chooses a chain from the cluster to ensure balanced sampling. We crop structures into patches containing 128 residues by first choosing a seed residue, and then selecting its 127 nearest neighbors based on C-beta distances. To simulate mutations, we masked the rotamers of 10% of residues in the patch, and we added noise to the rotamers of residues whose C-beta distance to the closest masked residue was less than 8Å.

The SKEMPI2 database (Jankauskaitė et al., 2019) is used to train the models for $\Delta\Delta G$ prediction described in Section 3.4. We split the dataset into 3 folds by structure, each containing unique protein complexes that do not appear in other folds. Two folds are used for training and validation, and the remaining fold is used for testing. This approach yields 3 different sets of parameters and ensures that every data point in SKEMPI2 is tested once.

## 4 RESULTS

### 4.1 PREDICTION OF THE EFFECT OF MUTATIONS ON BINDING

**Baselines** We evaluate the performance of our two $\Delta\Delta G$ predictors, **RDE-Linear** and **RDE-Network**, against several categories of baseline methods. The first category comprises traditional empirical energy functions, including **Rosetta** Cartesian ddG (Park et al., 2016; Alford et al., 2017; Leman et al., 2020) and **FoldX** (Delgado et al., 2019). The second category consists of sequence/evolution-based methods, represented by **ESM-1v** (Meier et al., 2021), **PSSM** (position-specific scoring matrix), **MSA Transformer** (Rao et al., 2021), and **Tranception** (Notin et al., 2022). The third category includes end-to-end learning models, such as **DDGPred** (Shan et al., 2022), and a model that shares the same encoder architecture as RDE but uses an MLP to directly predict $\Delta\Delta G$ (**End-to-End**). The fourth category consists of unsupervised/semi-supervised learning methods, including **ESM-IF** (Hsu et al., 2022) and Masked Inverse Folding (MIF) (Yang et al., 2022). Similar to our RDE-Network, this class of methods pre-train a network on structures and use the pretrained representations to predict $\Delta\Delta G$s. The baseline MIF network also uses the same encoder architecture as RDE for comparison. There are two variants for $\Delta\Delta G$ prediction: **MIF-$\Delta$logit**, which uses the difference in log-probability of amino acid types to predict $\Delta\Delta G$, and **MIF-Network**, which predicts $\Delta\Delta G$ from the learned representations using the same network architecture as RDE-Network. Finally, given that our method is based on conformational flexibility, we train a network to predict the **B-factor** of residues and use predicted B-factors in place of entropy to predict $\Delta\Delta G$.

**Metrics** We use five metrics to evaluate the accuracy of $\Delta\Delta G$ prediction: **Pearson** correlation coefficient, **Spearman**'s rank correlation coefficient, minimized **RMSE** (root mean squared error), minimized **MAE** (mean absolute error), and **AUROC** (area under the receiver operating characteristic). To calculate AUROC, mutations are classified based on the sign of $\Delta\Delta G$. In practical applications, the correlation for one specific protein complex is often of greater interest. Therefore, we group mutations by structure, discard groups with less than 10 mutation data points and calculate correlations for each structure separately. This leads to two additional metrics: average **per-structure Pearson** correlation coefficient and average **per-structure Spearman** correlation coefficient.

**Results** According to Table 1, our RDE-Network outperforms all the baselines. Notably, it demonstrates a significant improvement in per-structure correlations, indicating its greater reliability for practical applications. The superior performance of RDE-Network over MIF-Network suggests that representations derived from fitting rotamer densities are more effective than those from masked in-

Table 1: Evaluation of $\Delta\Delta G$ prediction on the SKEMPI2 dataset. RDE-Network outperforms baseline methods. Most notably, RDE-Network significantly improves per-structure correlations, which are more relevant to practical applications.

| Category | Method | Per-Structure | | Overall | | | | |
| | | Pearson | Spearman | Pearson | Spearman | RMSE | MAE | AUROC |
| --- | --- | --- | --- | --- | --- | --- | --- | --- |
| Sequence Based | ESM-1v | 0.0073 | -0.0118 | 0.1921 | 0.1572 | 1.9609 | 1.3683 | 0.5414 |
| | PSSM | 0.0826 | 0.0822 | 0.0159 | 0.0666 | 1.9978 | 1.3895 | 0.5260 |
| | MSA Transf. | 0.1031 | 0.0868 | 0.1173 | 0.1313 | 1.9835 | 1.3816 | 0.5768 |
| | Tranception | 0.1348 | 0.1236 | 0.1141 | 0.1402 | 2.0382 | 1.3883 | 0.5885 |
| Energy Function | Rosetta | 0.3284 | 0.2988 | 0.3113 | 0.3468 | 1.6173 | 1.1311 | 0.6562 |
| | FoldX | 0.3789 | 0.3693 | 0.3120 | 0.4071 | 1.9080 | 1.3089 | 0.6582 |
| Supervised | DDGPred | 0.3750 | 0.3407 | **0.6580** | 0.4687 | **1.4998** | **1.0821** | 0.6992 |
| | End-to-End | 0.3873 | 0.3587 | 0.6373 | 0.4882 | 1.6198 | 1.1761 | 0.7172 |
| Unsup. / Semisup. | B-factor | 0.2042 | 0.1686 | 0.2390 | 0.2625 | 2.0411 | 1.4402 | 0.6044 |
| | ESM-IF | 0.2241 | 0.2019 | 0.3194 | 0.2806 | 1.8860 | 1.2857 | 0.5899 |
| | MIF-$\Delta$logit | 0.1585 | 0.1166 | 0.2918 | 0.2192 | 1.9092 | 1.3301 | 0.5749 |
| | MIF-Net. | 0.3965 | 0.3509 | 0.6523 | 0.5134 | 1.5932 | 1.1469 | 0.7329 |
| Ours | RDE-Linear | 0.2903 | 0.2632 | 0.4185 | 0.3514 | 1.7832 | 1.2159 | 0.6059 |
| | RDE-Net. | **0.4448** | **0.4010** | 0.6447 | **0.5584** | 1.5799 | 1.1123 | **0.7454** |

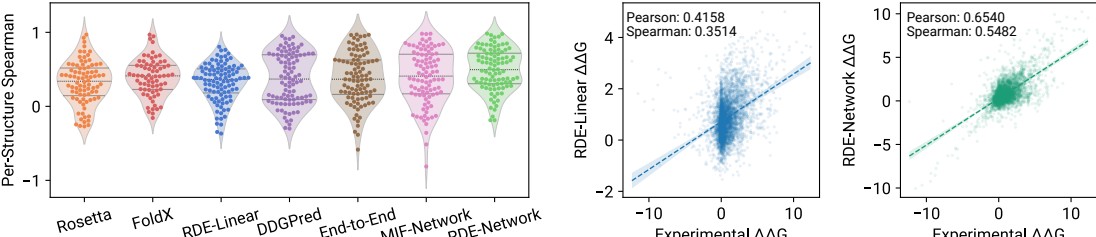

Figure 3: **Left**: The distribution of per-structure Spearman correlation coefficients. **Middle**: Correlation between experimental $\Delta\Delta G$s and $\Delta\Delta G$s predicted by RDE-Linear. **Right**: Correlation between experimental $\Delta\Delta G$s and $\Delta\Delta G$s predicted by RDE-Network.

verse folding, as RDE captures atomic interactions well by modeling the conformation of sidechain atoms.

RDE-Linear achieves comparable performance to Rosetta and outperforms some unsupervised learning baselines. While it does not surpass most baseline methods over the entire SKEMPI2 dataset, we observe that its performance is better when considering only single-point mutations (Table 6 in the appendix). This might be attributed to the fact that simple linear models cannot capture well the non-linear relationship dominating multi-point mutations. Nevertheless, RDE-Linear demonstrates that using the basic statistics of the estimated rotamer density alone can predict $\Delta\Delta G$, which lays the foundation for the more accurate RDE-Network.

Sequence-based models do not accurately predict $\Delta\Delta G$ for protein-protein binding, as discussed in Section 2.2. Figure 3 shows the distribution of per-complex correlation coefficients. Please refer to Section C of the appendix for more results and discussion.

## 4.2 OPTIMIZATION OF HUMAN ANTIBODIES AGAINST SARS-CoV-2

In Shan et al. (2022), the authors report five single-point mutations on a human antibody against SARS-CoV-2 that enhance neutralization effectiveness. These mutations are among the 494 possible single-point mutations on the heavy chain CDR region of the antibody. We use the most competitive methods benchmarked in Section 4.1 to predict $\Delta\Delta G$s for all the single-point mutations and rank them in ascending order (lowest $\Delta\Delta G$ in the top). The effectiveness of a predictor is determined by the number of favorable mutations ranked in the top place. As shown in Table 2, RDE-Network and DDGPred successfully identify three mutations (Ranking $\leq 10\%$), with RDE-Network ranking them higher.

Table 2: Rankings of the five favorable mutations on the human antibody against SARS-CoV-2 by various competitive methods. RDE-Network ranks 3 of the 5 mutations in the top place ($<10\%$).

| Method | TH31W | AH53F | NH57L | RH103M | LH104F |
|---|---|---|---|---|---|
| Rosetta | 10.73% | 76.72% | 93.93% | 11.34% | 27.94% |
| FoldX | 13.56% | **6.88%** | **5.67%** | 16.60% | 66.19% |
| DDGPred | 68.22% | **2.63%** | 12.35% | **8.30%** | **8.50%** |
| End-to-End | 29.96% | **2.02%** | 14.17% | 52.43% | 17.21% |
| MIF-Net. | 24.49% | **4.05%** | **6.48%** | 80.36% | 36.23% |
| RDE-Net. | **1.62%** | **2.02%** | 20.65% | 61.54% | **5.47%** |

Table 3: Linear regression shows that the relevant terms estimated by RDE correlate significantly to $\Delta\Delta G$.

| Var. | Sign | Coef. | P-value | Signif. |
|---|---|---|---|---|
| $H^{\text{bound}}_{\mathcal{ML}}$ | + | 0.5122 | <0.001 | *** |
| $H^{\text{bound}}_{\mathcal{MR}}$ | + | 0.1808 | 0.005 | ** |
| $H^{\text{unbnd}}_{\mathcal{ML}}$ | - | 0.2849 | <0.001 | *** |
| $H^{\text{bound}}_{\mathcal{WL}}$ | - | 0.4939 | <0.001 | *** |
| $H^{\text{bound}}_{\mathcal{WR}}$ | - | 0.2515 | <0.001 | *** |
| $H^{\text{unbnd}}_{\mathcal{WL}}$ | + | 0.2471 | <0.001 | *** |
| $H^{\text{unbnd}}_{R}$ | / | 0.0325 | 0.230 | - |
| Bias | / | 0.1888 | <0.001 | *** |

### 4.3 ANALYSIS OF THE ENTROPY ESTIMATED BY ROTAMER DENSITY ESTIMATOR

**Statistical Significance** To demonstrate a statistically significant relationship between the entropy estimated by RDE and experimental $\Delta\Delta G$ values, we conduct linear regression analysis using the RDE-Linear model defined in Eq. 9. The linear model consists of seven coefficients and one bias term: $w^{\text{bound}}_{\mathcal{WL}}$, $w^{\text{bound}}_{\mathcal{WR}}$, $w^{\text{unbnd}}_{\mathcal{WL}}$, $w^{\text{bound}}_{\mathcal{ML}}$, $w^{\text{bound}}_{\mathcal{MR}}$, $w^{\text{unbnd}}_{\mathcal{ML}}$, $w^{\text{unbnd}}_{R} = (w^{\text{unbnd}}_{\mathcal{MR}} - w^{\text{unbnd}}_{\mathcal{WR}})$, and $b$. Note that $w^{\text{unbnd}}_{\mathcal{MR}}$ and $w^{\text{unbnd}}_{\mathcal{WR}}$ are merged into $w^{\text{unbnd}}_{R}$, as the receptor is not mutated. We perform linear regression on the SKEMPI2 single-mutation dataset and present the regression coefficients, bias, and P-values in Table 3. According to the statistics, all entropy terms, except the entropy of unbound receptor $H^{\text{unbnd}}_{R}$ (coefficient $w^{\text{unbnd}}_{R}$), show a significant relationship with experimental $\Delta\Delta G$s. The coefficients of the significant terms are all positive and roughly similar. The entropy of the unbound receptor $H^{\text{unbnd}}_{R}$ has no contribution because the receptor alone does not involve in the mutation. These results agree with the thermodynamic definition of the change in binding free energy: $\Delta\Delta G = \Delta G^{\mathcal{M}} - \Delta G^{\mathcal{W}} = (G^{\mathcal{M}}_{LR} - G^{\mathcal{M}}_{L} - G^{\mathcal{M}}_{R}) - (G^{\mathcal{W}}_{LR} - G^{\mathcal{W}}_{L} - G^{\mathcal{W}}_{R})$, where $G^{\mathcal{M}}_{R}$ and $G^{\mathcal{W}}_{R}$ cancel each other out as the receptor is unmutated, therefore indicating that our model captures well the thermodynamics underlying protein-protein interactions. For a detailed discussion about the thermodynamic background, please refer to Section B in the appendix.

**Correlation Between Estimated Entropy and B-factors** B-factor is an experimental measurement that quantifies the conformational flexibility. We calculate the average b-factor of sidechain atoms of residues in the test split of the PDB-REDO dataset. Then, we estimate the conformational entropy for each residue in the test split using RDE. The average Pearson correlation coefficient between these two quantities is 0.4637, and the average Spearman coefficient is 0.4282. Detailed results are presented in Table 8 in the appendix. In summary, this indicates that there is a correlation between the entropy estimated by RDE and experimentally determined conformational flexibility measured by B-factor.

### 4.4 PREDICTION OF SIDECHAIN CONFORMATIONS

RDE is a generative model for protein sidechain structures, which can predict sidechain conformations by sampling from the estimated distribution. We use RDE to sample sidechain torsional angles (rotamers) for structures with 10% sidechains removed in our test split of PDB-REDO. For each residue, 10 rotamers are sampled independently, and the one with the highest probability is selected as the

Table 4: Mean absolute error of the predicted sidechain torsional angles.

| | $\chi_1$ | $\chi_2$ | $\chi_3$ | $\chi_4$ |
|---|---|---|---|---|
| SCWRL4 | 24.33° | 32.84° | 47.42° | 56.15° |
| Rosetta | 23.98° | 32.14° | 48.49° | 58.78° |
| RDE | **16.02°** | **28.75°** | **45.38°** | **53.26°** |

final prediction. We compare RDE with two baseline methods Rosetta (fixbb) (Leman et al., 2020) and SCWRL4 (Krivov et al., 2009). Our results shown in Table 4 demonstrate that RDE outperforms the baselines on all four torsional angles in terms of angular errors. For a detailed per-amino-acid accuracy, please refer to Table 9 in the appendix.

## 5 CONCLUSIONS

In this work, we introduce Rotamer Density Estimator (RDE) which estimates the distribution of rotamers for protein sidechains. We demonstrate that RDE leads to improved accuracy in predicting binding $\Delta\Delta G$ compared to other methods. One limitation of RDE is the inability to model backbone flexibility directly which is an important future direction for extending the proposed model. Nonetheless, our work highlights the potential of using machine learning techniques to improve mutational effect prediction for protein-protein interaction.

ACKNOWLEDGMENTS

Supported by National Key R&D Program of China No. 2021YFF1201600.

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

# A  IMPLEMENTATION DETAILS

## A.1  NETWORK ARCHITECTURE

The encoder of the rotamer density estimator is a stack of attention layers invariant to the rotation and translation (Jumper et al., 2021) of the input protein structure. Let $\boldsymbol{h}_i^\ell$ denote the embedding of the $i$-th amino acid output by the previous attention layer. The logit of attention weights between residue $i$ and residue $j$ is defined as:

$$a_{ij} = \langle Q(\boldsymbol{h}_i^\ell), K(\boldsymbol{h}_i^\ell) \rangle + G(\boldsymbol{z}_{ij}) + H\left(\{\boldsymbol{O}_i^\intercal(\boldsymbol{x}_j^k - \boldsymbol{x}_i^k)\}_k\right), \tag{10}$$

where $Q, K, G$, and $H$ are MLP networks that transform the embeddings into queries, keys, pairwise bias, and distance bias, respectively. The attention weight is computed by taking the softmax on the $j$ dimension: $w_{ij} = \mathrm{softmax}_j(a_{ij})$. In practice, we use multiple attention heads. Each attention head has different attention weights. The vector to update the representation of residue $i$ from residue $j$ is computed by:

$$\boldsymbol{v}_{ij} = V\left(\boldsymbol{h}_j^\ell, \boldsymbol{z}_{ij}, \{\boldsymbol{O}_i^\intercal(\boldsymbol{x}_j^k - \boldsymbol{x}_i^k)\}_k\right). \tag{11}$$

Finally, the sum of $\{\boldsymbol{v}_{ij}\}_j$ weighted by $\{w_{ij}\}_j$ is used to update the representation of residue $j$ using residual connection and layer normalization, similar to the standard transformer architecture.

The rotamer density estimator has 6 encoder layers. Node features and pairwise features have 128 channels and 64 channels respectively. The normalizing flow has 8 blocks and each spline has 65 knots, dividing $[0, 2\pi]$ into 64 bins.

## A.2  TRAINING

The rotamer density estimator is trained using the Adam optimizer for 200K iterations. The initial learning rate is 0.0001. The learning rate decays by 0.8 if the validation loss does not decrease in the last 5 validation steps (the model is validated every 1000 iterations), until the learning rate reaches 0.000001. The batch size is 64. It takes 8h56m in total on a single A100 GPU.

To emulate mutations, the rotamers of 10% of amino acids are masked. Noise is added to the rotamers of amino acids whose C-beta distance to the closest masked amino acid is less than 8.0Å. The noise added to $\chi$ angles consists of two components. The first component is Gaussian noise centered at 0 wrapped into $[-\pi, \pi]$. It standard deviation is dependent on the C-beta distance: $\sigma_{ij} = -\frac{1}{16}\beta_{ij} + 1$, where $i$ is the index of the nearest masked amino of the $j$-th amino acid that the noise is added to, and $\beta_{ij}$ is the C-beta distance between them. The second component is random flipping (adding $\pi$ to the angle). Every $\chi$ angle in the 8Å neighborhood has 25% chance of being flipped. Our noise model is totally empirical. There are other ways to perturb rotamers, for example, using rotamer libraries (Dunbrack Jr & Karplus, 1993; Bower et al., 1997; Dunbrack Jr, 2002; Shapovalov & Dunbrack Jr, 2011). We leave the problem of finding an optimal noise model that emulates mutations in future work.

## A.3  BASELINES

Baselines that require training and calibration using the SKEMPI2 dataset (DDGPred, End-to-End, B-factor, MIF-$\Delta$logit, MIF-Network) are trained independently using the 3 different splits of the dataset as described in Section 3.5. This is to ensure that every data point in the SKEMPI2 dataset is tested on once. Below are descriptions of the implementation of baseline methods.

**Rosetta** (Alford et al., 2017; Leman et al., 2020) The version we used is 2021.16, and the scoring function is `ref2015_cart`. Every protein structures in the SKEMPI2 dataset are first pre-processed using the `relax` application. The mutant structure is built by `cartesian_ddg`. The binding free energies of both wild-type and mutant structures are predicted by `interface_energy` (dG_separated/dSASAx100). Finally, the binding $\Delta\Delta G$ is calculated by substracting the binding energy of the wild-type structure from the binding energy of the mutant.

**FoldX** (Delgado et al., 2019) Structures are first relaxed by the `RepairPDB` command. Mutant structures are built with the `BuildModel` command based on the repaired structure. The change in binding free energy $\Delta\Delta G$ is calculated by subtracting the wild-type energy from the mutant energy.

**ESM-1v** (Meier et al., 2021) We use the implementation provided in the ESM open-source repository. Protein language models can only predict the effect of mutations for single protein sequences. Therefore, we ignore cases where mutations are on multiple sequences. We extract the sequence of the mutated protein chain from the `SEQRES` entry of the PDB file. We use `masked-marginals` mode to score both wild-type and mutant sequences and use their difference as the estimation of $\Delta\Delta G$.

**PSSM** We construct MSAs from the Uniref90 database (Suzek et al., 2007) for chains with mutation annotations in the SKEMPI dataset. We use Jackhmmer (Johnson et al., 2010) version 3.3.1 following the setting in Meier et al. (2021). The MSAs are filtered using HHfilter (Steinegger et al., 2019) with coverage 75 and sequence identity 90. This HHfilter parameter is reported to have the best performance for MSA Transformer according to Meier et al. (2021). We calculate position-specific scoring matrices (PSSM) and use the change in probability as the prediction of $\Delta\Delta G$.

**MSA Transformer** (Rao et al., 2021) We use the implementation provided in the ESM open-source repository. We input the MSAs constructed during the evaluation of PSSM to MSA Transformer. We used `masked-marginals` mode to score both wild-type and mutant sequences and use their difference as the prediction of $\Delta\Delta G$.

**Tranception** (Notin et al., 2022) We use the implementation provided in the Tranception open-source repository. We predict mutation effects using the large model checkpoint. Previously built MSAs (not filtered by HHfilter) are used for inference-time retrieval.

**DDGPred** (Shan et al., 2022) We use the implementation accompanying the paper by Shan et al. (2022). Since this model requires predicted sidechain structures of the mutant, we use mutant structures packed during our evaluation of Rosetta to train the model and run prediction.

**End-to-End** The end-to-end model shares the same encoder architecture as the rotamer density estimator. The difference is that in the RDE normalizing flows follow the encoder to model rotamer distributions, but in the end-to-end model, the embeddings are directly fed to an MLP to predict $\Delta\Delta G$.

**B-factor** This model predicts per-atom b-factors for proteins. It has the same encoder architecture as the RDE. Following the encoder is an MLP that predicts a vector for each amino acid where each dimension is the predicted b-factor of different atoms in the amino acid. The amino acid-level b-factor is calculated by averaging the atom-level b-factors. The predicted b-factors are used as a measurement of conformational flexibility. They are used to predict $\Delta\Delta G$ using the linear model same as RDE-Linear defined in Eq.9.

**ESM-IF** (Hsu et al., 2022) ESM-IF can score protein sequences using the log-likelihood. The scoring function implementation is provided in the ESM repository. We enable the `--multichain_backbone` flag to let the model see the whole protein-protein complex. We subtract the log-likelihood of the wild-type from the mutant to predict $\Delta\Delta G$.

**MIF Architecture** The masked inverse folding (MIF) network uses the same encoder architecture as RDE. Following the encoder is a per-amino-acid 20-category classifier that predicts the type of masked amino acids. We use the same PDB-REDO train-test split to train the model. At training time, we randomly crop a patch consisting of 128 residues, and randomly mask 10% amino acids. The model learns to recover the type of masked amino acids with the standard cross entropy loss.

**MIF-$\Delta$logit** To score mutations, we first mask the type of mutated amino acids. Then, we use the log probability of the amino acid type as the score. Analogously, we have the score of the wild-type bound ligand, wild-type bound receptor, wild-type unbound ligand, unbound receptor, mutated bound ligand, mutated bound receptor, and mutated unbound ligand. Therefore, we use the linear model identical to RDE-Linear (Eq.9) to predict $\Delta\Delta G$ from the scores.

**MIF-Network** This is similar to RDE-Network. The difference is that we use the pre-trained encoder of MIF rather than the encoder of RDE. We also freeze the MIF encoder as we aim to utilize the unsupervised representations.

## A.4 SOURCE CODE

Available at https://github.com/luost26/RDE-PPI.

# B  BACKGROUND

This section introduces the thermodynamic principle underlying the design of RDE, which connects entropy and binding affinity.

The Gibbs free energy of *association* is the physical quantity used to measure the binding affinity between two groups of protein:

$$\Delta G_a = \Delta H - T\Delta S. \tag{12}$$

In this equation, $\Delta H$ is the change in enthalpy upon the formation of the complex, which is generally assumed to be negligible as no covalent bonds are formed or broken upon protein-protein binding. $T$ is the temperature parameter, and $\Delta S$ is the change in entropy upon binding (Kastritis & Bonvin, 2013). By ignoring $\Delta H$ and expanding $\Delta S$, we can rewrite $\Delta G_a$ as $\Delta G_a = T(S_d - S_a)$, where $S_a$ is the entropy of the proteins in the bound state (complex), and $S_d$ is the entropy in the unbound state (separated). To predict binding affinity, we need to calculate the entropy of the separated proteins ($S_d$) and the entropy of the protein complex ($S_a$).

The entropy $S$ is defined by the Boltzmann expression:

$$S = -k_B \int p(\boldsymbol{x}) \log p(\boldsymbol{x}) \mathrm{d}\boldsymbol{x} = -k_B \mathbb{E}_{\boldsymbol{x} \sim p} \log p(\boldsymbol{x}), \tag{13}$$

where $p(\boldsymbol{x})$ is the distribution of conformation $\boldsymbol{x}$ and $k_B$ is the Boltzmann constant (Brady & Sharp, 1997). To calculate the entropy of the protein complex $S_a$, we need to evaluate the integral with respect to the distribution of the complex conformation denoted by $p_a$. To calculate the entropy of the separated proteins $S_d$, we can factorize the probability density $p_d = p_{\text{ligand}}(\boldsymbol{x}_{\text{ligand}}) \cdot p_{\text{receptor}}(\boldsymbol{x}_{\text{receptor}})$ and then evaluate $-k_B \left( \mathbb{E}_{\boldsymbol{x} \sim p_{\text{ligand}}} \log p_{\text{ligand}}(\boldsymbol{x}) + \mathbb{E}_{\boldsymbol{x} \sim p_{\text{receptor}}} \log p_{\text{receptor}}(\boldsymbol{x}) \right)$. Substituting $S_d$ and $S_a$ in $\Delta G_a$ with these expressions, we can obtain the formula:

$$\Delta G_a = -k_B T \left( \mathbb{E}_{\boldsymbol{x} \sim p_{\text{ligand}}} \log p_{\text{ligand}}(\boldsymbol{x}) + \mathbb{E}_{\boldsymbol{x} \sim p_{\text{receptor}}} \log p_{\text{receptor}}(\boldsymbol{x}) - \mathbb{E}_{\boldsymbol{x} \sim p_a} \log p_{\text{a}}(\boldsymbol{x}) \right), \tag{14}$$

which indicates that we can predict binding affinity by estimating the entropy of the conformation distributions of the protein complex $p_a$ and each separated protein $p_{\text{ligand}}, p_{\text{receptor}}$.

We assume that sidechain conformation changes are the major determinant of protein-protein binding, so we can keep the protein backbone fixed and model only the distribution of sidechain conformations (rotamers) (Najmanovich et al., 2000; Cole & Warwicker, 2002). This assumption leads to the core component of this work, Rotamer Density Estimator (RDE), which approximates $p_a, p_{\text{ligand}}$, and $p_{\text{receptor}}$, enabling us to estimate $\Delta G_a$ by evaluating the entropy of these distributions.

Finally, to evaluate the effect of mutations, we apply RDE to estimate $G_a$ for both the wild-type and the mutant. We then calculate the difference between the $G_a$ values of the mutant and wild-type, yielding the quantity $\Delta\Delta G$:

$$\Delta\Delta G = \Delta G_{\text{mutant}} - \Delta G_{\text{wild-type}}. \tag{15}$$

We refer the reader to Brady & Sharp (1997) and Kastritis & Bonvin (2013) for a comprehensive treatment of the relationship between binding affinity and entropy.

## C  ADDITIONAL RESULTS

### C.1  FURTHER ANALYSIS ON THE PERFORMANCE OF SEQUENCE-BASED METHODS

As discussed in Section 2.2, sequence-based (evolution-based) methods are unsuitable for predicting protein-protein interactions due to the lack of co-evolutionary information between the two proteins. This is supported by the results in Table 1, which indicate that sequence-based methods are inaccurate in predicting $\Delta\Delta G$. We analyzed two classes of PPIs to better understand the performance of sequence-based methods on PPIs.

Antibody-antigen binding is a typical class of PPI that lacks co-evolutionary information. The variability of the binding interface of antibodies (complementarity determining region, CDR) means that there is no evolutionary history in the region, making it infeasible to mine the preference of mutations from sequence databases. Additionally, in most cases, antigens do not evolve to increase binding to specific antibodies, so sequence databases provide little information about mutational effects on antibody-antigen binding. We evaluated per-structure Spearman correlation coefficients of MSA Transformer and RDE-Network on antibody-antigen complexes from the SKEMPI datasets. The average Spearman score of MSA Transformer is 0.0744 and the average score of RDE-Network is 0.4284. Figure 4 shows the results, where the $x$-axis and $y$-axis are the per-structure Spearman coefficients of MSA Transformer and RDE-Network respectively. Orange crosses represent antibody-antigen complexes, and blue dots represent other complexes. The results indicate that when we have little co-evolutionary information such as in antibody-antigen binding, structure-based methods, represented by our RDE, perform better than evolution-based methods, represented by MSA Transformer.

When the proteins in a complex come from the same organism, evolution-based methods are more likely to be effective. These proteins usually function together in the organism, so they evolve together. Mutations that enhance the complexation may be more favorable, and this preference might be reflected in evolutionary history. We inspected 10 complexes on which MSA Transformer performed the best in terms of Spearman coefficients and found that 9 of them consist of proteins from a single organism (Table 5). However, when evaluating MSA Transformer on all the single-organism complexes, its Spearman score is 0.1651, which is still low. The reason may be that even if the proteins in a complex come from the same organism, the member protein might also bind to other proteins to be functional. In this case, it is more challenging to predict its binding in a specific complex according to its general evolutionary history.

Figure 4: Per-structure Spearman correlation coefficients of the prediction by MSA Transformer and RDE-Network. Orange crosses represent antibody-antigen complexes and blue dots represent other types of complexes. Axes are cropped to $[0, 1]$.

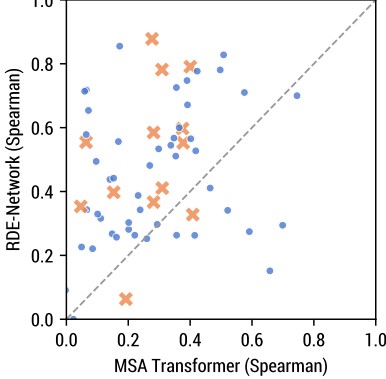

Table 5: Protein complexes on which MSA Transformer performs the best in terms of Spearman coefficients.

| Complex | MSA Transf. Spearman | Number of Organisms |
|---|---|---|
| 4OFY_A_D | 0.7454 | 1 |
| 2J0T_A_D | 0.6990 | 1 |
| 1AK4_A_D | 0.6578 | 2 |
| 2SIC_E_I | 0.5915 | 1 |
| 1BRS_A_D | 0.5755 | 1 |
| 3SE3_B_A | 0.5213 | 1 |
| 1EMV_A_B | 0.5082 | 1 |
| 1B41_A_B | 0.4972 | 1 |
| 1Z7X_W_X | 0.4647 | 1 |
| 3SE4_B_C | 0.4268 | 1 |

## C.2 Single-Mutation and Multi-Mutation

Table 6: Evaluation of $\Delta\Delta G$ predictors on the single-mutation subset of SKEMPI2.

| Method | Per-Struct. Pearson | Per-Struct. Spearman | Overall Pearson | Overall Spearman | RMSE (kcal/mol) | MAE (kcal/mol) | AUROC |
|---|---|---|---|---|---|---|---|
| ESM-1v | 0.0422 | 0.0273 | 0.1914 | 0.1572 | 1.7226 | 1.1917 | 0.5492 |
| PSSM | 0.1215 | 0.1229 | 0.1224 | 0.0997 | 1.7420 | 1.2055 | 0.5659 |
| MSA Transf. | 0.1415 | 0.1293 | 0.1755 | 0.1749 | 1.7294 | 1.1942 | 0.5917 |
| Tranception | 0.1912 | 0.1816 | 0.1871 | 0.1987 | 1.7455 | 1.1708 | 0.6089 |
| Rosetta | 0.3284 | 0.2988 | 0.3113 | 0.3468 | 1.6173 | 1.1311 | 0.6562 |
| FoldX | 0.3908 | 0.3640 | 0.3560 | 0.3511 | 1.5576 | 1.0713 | 0.6478 |
| DDGPred | 0.3711 | 0.3427 | 0.6515 | 0.4390 | 1.3285 | 0.9618 | 0.6858 |
| End-to-End | 0.3818 | 0.3426 | 0.6605 | 0.4594 | 1.3148 | 0.9569 | 0.7019 |
| B-factor | 0.1884 | 0.1661 | 0.1748 | 0.2054 | 1.7242 | 1.1889 | 0.6100 |
| ESM-IF | 0.2308 | 0.2090 | 0.2957 | 0.2866 | 1.6728 | 1.1372 | 0.6051 |
| MIF-$\Delta$logit | 0.1616 | 0.1231 | 0.2548 | 0.1927 | 1.6928 | 1.1671 | 0.5630 |
| MIF-Net. | 0.3952 | 0.3479 | **0.6667** | 0.4802 | **1.3052** | 0.9411 | 0.7175 |
| RDE-Linear | 0.3192 | 0.2837 | 0.3796 | 0.3394 | 1.5997 | 1.0805 | 0.6027 |
| RDE-Net. | **0.4687** | **0.4333** | 0.6421 | **0.5271** | 1.3333 | **0.9392** | **0.7367** |

Table 7: Evaluation of $\Delta\Delta G$ predictors on the multi-mutation subset of SKEMPI2.

| Method | Per-Struct. Pearson | Per-Struct. Spearman | Overall Pearson | Overall Spearman | RMSE (kcal/mol) | MAE (kcal/mol) | AUROC |
|---|---|---|---|---|---|---|---|
| ESM-1v | -0.0599 | -0.1284 | 0.1923 | 0.1749 | 2.7586 | 2.1193 | 0.5415 |
| PSSM | -0.0174 | -0.0504 | -0.1126 | -0.0458 | 2.7937 | 2.1499 | 0.4442 |
| MSA Transf. | -0.0097 | -0.0400 | 0.0067 | 0.0030 | 2.8115 | 2.1591 | 0.4870 |
| Tranception | -0.0688 | -0.0120 | -0.0185 | -0.0184 | 2.9280 | 2.2359 | 0.4874 |
| Rosetta | 0.1915 | 0.0836 | 0.1991 | 0.2303 | 2.6581 | 2.0246 | 0.6207 |
| FoldX | 0.2801 | 0.2771 | 0.2347 | 0.4137 | 2.5290 | 1.8639 | 0.6828 |
| DDGPred | 0.3912 | 0.3896 | 0.5938 | 0.5150 | 2.1813 | 1.6699 | 0.7590 |
| End-to-End | 0.4178 | 0.4034 | 0.5858 | 0.4942 | 2.1971 | 1.7087 | 0.7532 |
| B-factor | 0.2078 | 0.1850 | 0.2009 | 0.2445 | 2.6557 | 2.0186 | 0.5876 |
| ESM-IF | 0.2016 | 0.1491 | 0.3260 | 0.3353 | 2.6446 | 1.9555 | 0.6373 |
| MIF-$\Delta$logit | 0.1053 | 0.0783 | 0.3358 | 0.2886 | 2.5361 | 1.8967 | 0.6066 |
| MIF-Net. | 0.3968 | 0.3789 | 0.6139 | 0.5370 | 2.1399 | 1.6422 | **0.7735** |
| RDE-Linear | 0.1763 | 0.2056 | 0.4583 | 0.4247 | 2.4460 | 1.8128 | 0.6573 |
| RDE-Net. | **0.4233** | **0.3926** | **0.6288** | **0.5900** | **2.0980** | **1.5747** | 0.7749 |

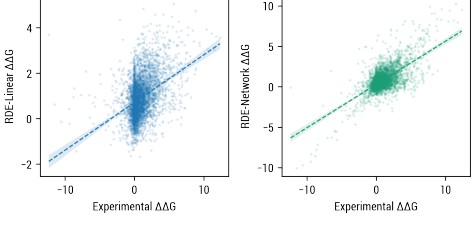

Figure 5: Correlation between experimental $\Delta\Delta G$s and $\Delta\Delta G$s predicted by RDE-Linear and RDE-Network on SKEMPI2 single-mutation subset.

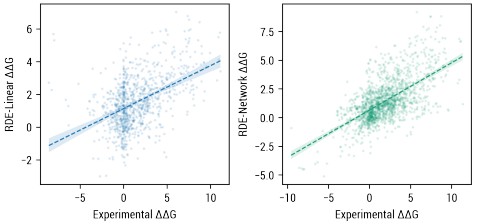

Figure 6: Correlation between experimental $\Delta\Delta G$s and $\Delta\Delta G$s predicted by RDE-Linear and RDE-Network on SKEMPI2 multi-mutation subset.

## C.3 CORRELATION WITH B-FACTOR

Table 8: Correlation between average sidechain B-factors and entropy estimated by RDE. Scatter plots of each amino acid type are in Figure 7.

| Type | Pearson | Spearman |
|------|---------|----------|
| ARG | 0.4683 | 0.4904 |
| ASN | 0.4329 | 0.3826 |
| ASP | 0.4680 | 0.4408 |
| CYS | 0.4099 | 0.4099 |
| GLN | 0.4691 | 0.4463 |
| GLU | 0.4683 | 0.4861 |
| HIS | 0.3743 | 0.3261 |
| ILE | 0.5288 | 0.4677 |
| LEU | 0.5281 | 0.4638 |
| LYS | 0.5479 | 0.5777 |
| MET | 0.5026 | 0.4758 |
| PHE | 0.3615 | 0.2667 |
| SER | 0.5368 | 0.5247 |
| THR | 0.5945 | 0.5263 |
| TRP | 0.3263 | 0.2910 |
| TYR | 0.3588 | 0.2493 |
| VAL | 0.5068 | 0.4536 |
| Average | 0.4637 | 0.4282 |

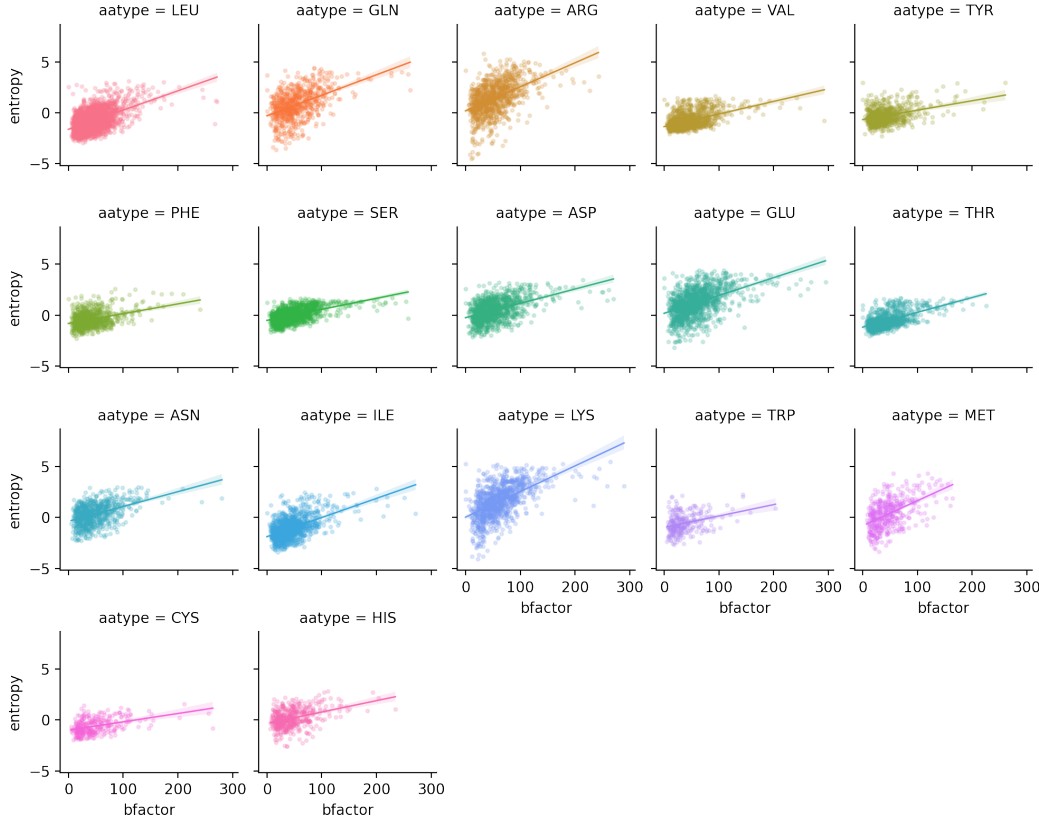

Figure 7: Scatter plots of average sidechain B-factors and estimated entropy.

## C.4 SIDECHAIN CONFORMATION PREDICTION

Table 9: Mean absolute error of the predicted sidechain torsional angles.

| Type | $\chi$ | SCWRL4 | Rosetta | RDE | Type | $\chi$ | SCWRL4 | Rosetta | RDE |
|------|--------|--------|---------|-----|------|--------|--------|---------|-----|
| ARG | 1 | 29.30 | 30.50 | **20.90** | LEU | 1 | 13.97 | 14.15 | **9.41** |
| | 2 | **28.68** | 36.12 | 28.72 | | 2 | 23.76 | 27.61 | **19.32** |
| | 3 | 57.89 | 60.73 | **56.83** | LYS | 1 | 31.32 | 33.88 | **21.64** |
| | 4 | 60.35 | 63.62 | **56.76** | | 2 | **30.95** | 33.15 | 32.90 |
| ASN | 1 | 21.70 | 19.39 | **17.26** | | 3 | 38.90 | 42.07 | **37.70** |
| | 2 | 44.00 | 43.41 | **41.36** | | 4 | 51.94 | 53.94 | **49.76** |
| ASP | 1 | 25.75 | 22.62 | **17.54** | MET | 1 | 26.36 | 26.07 | **16.87** |
| | 2 | 23.90 | 21.26 | **21.15** | | 2 | 38.52 | 36.09 | **27.09** |
| CYS | 1 | 24.83 | 25.90 | **12.74** | | 3 | 55.11 | 58.77 | **50.21** |
| GLN | 1 | 33.16 | 31.53 | **22.78** | PHE | 1 | 12.30 | 13.08 | **9.22** |
| | 2 | 46.33 | **33.96** | 35.16 | | 2 | 12.40 | 12.35 | **9.45** |
| | 3 | 53.72 | 56.52 | **52.80** | SER | 1 | 47.19 | 46.83 | **25.66** |
| GLU | 1 | 35.45 | 34.69 | **26.14** | THR | 1 | 28.05 | 22.67 | **16.85** |
| | 2 | 38.23 | 38.43 | **34.60** | TRP | 1 | 14.52 | 18.64 | **8.50** |
| | 3 | 31.50 | 30.85 | **29.35** | | 2 | 31.74 | 31.44 | **24.42** |
| HIS | 1 | 23.15 | 19.12 | **17.88** | TYR | 1 | 11.39 | 14.56 | **9.25** |
| | 2 | 70.62 | **61.97** | 69.69 | | 2 | 11.37 | 14.45 | **8.35** |
| ILE | 1 | 13.92 | 14.65 | **8.86** | VAL | 1 | 21.31 | 19.41 | **10.82** |
| | 2 | 26.43 | 27.54 | **21.54** | | | | | |

