# OpenReview forum: "Rotamer Density Estimator is an Unsupervised Learner of the Effect of Mutations on Protein-Protein Interaction"
_ICLR.cc/2023/Conference — ICLR 2023 poster_

### Official Review · Reviewer_NQyq · 2022-10-21

**Confidence:** 3
**Correctness:** 3
**Technical Novelty And Significance:** 3
**Empirical Novelty And Significance:** 2
**Recommendation:** 5

**Clarity, Quality, Novelty And Reproducibility:**

Quality: Comparable (but not significantly improved?) empirical results to the state of the art for mutation effect prediction. The SKEMPI ddG prediction results lack some machine learning baselines (e.g. GVP, ESM-IF1).

Clarity: Two lingering questions: 1) Is there any rigorous biophysical justification for the entropy correlating with ddG? 2) Does the training data (crystallography) faithfully capture the distribution of conformations?

Originality: New approach to do density estimation for protein conformations.



**Strength And Weaknesses:**

Strengths:
+ Interesting approach for modeling the probability distribution of conformations with flow models.
+ The problem of modeling the distribution of conformation itself is of wide interest. (I would encourage the author to also explore more applications.)
+ Empirical results are based on a common benchmark dataset (SKEMPI).
+ Relatively clear presentation of the method and the results.

Weaknesses:
+ On the mutation effect prediction task evaluated in the paper, several machine learning baselines are missing (e.g. GVP, ESM-IF1).
+ Misleading bolding in the tables (when the proposed method is not the best, the best method is not bolded)
+ No confidence intervals for Table 1 and Table 2
+ Does the training data (crystallography) faithfully capture the distribution of conformations? Empirical proof or theoretical reasoning would be helpful.
+ Is there any rigorous biophysical justification for the entropy correlating with ddG?

**Summary Of The Paper:**

The authors proposed a new method (Rotamer Density Estimator): a flow-based generative model to estimate the probability distribution of conformation. The proposed Rotamer Density Estimator was evaluated on mutational effects for ddG (by using entropy of the probability distribution as the measure of flexibility).

**Summary Of The Review:**

Original approach for density estimation of protein conformations with limited improvement in empirical results.

---

> ### Author Response · Authors · 2022-11-13
> **Response (1/3)**
>
> We appreciate the reviewer’s valuable comment on our work! Below are our responses to the reviewer's concerns and clarification on some issues.
>
> ----------
>
> **[Clarification on the motivation of our work]**
>
> We have noticed that the reviewer might consider our work focuses on conformation generation. However, we would like to clarify that *generating conformations is **not** the primary purpose of the RDE*. The primary concern of our work is predicting the effect of mutations on PPIs. We observed the relationship between mutational effects and conformational flexibility. That’s why RDE has been derived — we need a tool to model conformational flexibility in order to predict mutational effects. *In short, conformation density estimation is not the end but rather a means to predict mutational effects*.
>
> Though, as the reviewer suggested, predicting conformations is a problem of broad interest, especially for large systems such as protein sidechains. Therefore, it would be valuable to explore applying the RDE (not necessarily flow-based, diffusion-based, and other potential variants are also viable) to conformation modeling in future work. At present, this direction is beyond the scope of this work as we focus on mutational effect prediction.
>
> ----------
>
> **[Q1] Additional baselines**
>
> As suggested by the reviewer, we have compared our method with ESM-IF and updated the results in Table 1. We evaluate ESM-IF using the official implementation for scoring protein sequences in multi-chain protein complexes. The implementation is taken from the ESM open-source repository.
>
> We find that both RDE-Linear and RDE-Network outperform ESM-IF. Although ESM-IF is trained using a significantly larger structure database, it does not explicitly model fine-grained atomic interactions between amino acids, which are critical in protein-protein interaction. It learns a coarse-level "language" of amino acid sequences conditional on the backbone structure. This could explain why RDE has better performance on PPIs.
>
> In addition to ESM-IF, we have also evaluated three sequence-based methods: PSSM, MSA Transformer, and Tranception, which require MSAs as input and are generally stronger than ESM-1v. Our finding is that the performance of these stronger methods are still very limited. This reinforces our argument in Section 2.2 --- sequence-based methods are not suitable for most PPIs due to the lack of co-evolutionary information. More details about the implementation and analysis of these MSA-based baselines are added to Section A.3 and Section B.1 in the appendix.
>
> ----------
>
> **[Q2] Formatting issues**
>
> We have fixed the bolding issues in Table 1 and Table 2. We have also rearranged Table 1, 6, and 7 for better clarity.
>
> ----------
>
> **[Q3] Confidence intervals**
>
> We have added confidence intervals of the metrics in Section B.3 of the appendix. Since we are unable to re-train some baselines (ESM, DDGPred), we cannot report error bars for them. We find that as the testset is large enough, the standard deviation is not large.
>
> ----------
>
> **[Q4] Improvement over previous methods**
>
> The most important application of PPI mutational effect predictors is to guide modulating the binding strength of *a specific* protein-protein complex (introduced in Section 1). That means biologists are interested in ranking mutations of a specific protein complex of interest, and it is generally not necessary to compare mutations that apply to different protein complexes.
>
> Therefore, to evaluate different methods' ability to rank mutations within a specific protein complex, we group data points by structure and calculate correlation coefficients separately for each structure (per-structure Pearson, per-structure Spearman). These per-structure correlation scores are more relevant to practical applications and should better reflect how good the models are for real-world challenges. According to Table 1, RDE-Network *surpasses all the baselines clearly in terms of the per-structure correlation scores*, which means that it is more practically reliable compared to previous methods.

---

> > ### Author Response · Authors · 2022-11-13
> > **Response (2/3)**
> >
> > **[Q5] Why does DDGPred seem to perform better in terms of RMSE, MAE, and Pearson?**
> >
> > This is because RMSE, MAE, and Pearson are sensitive to outlying ΔΔG values (whose absolute values are too large), so it overestimates the accuracy of DDGPred.
> >
> > To understand what happens, we give the following illustrative example:
> >
> > |                  | Mutation 1 | Mutation 2 | Mutation 3 | Mutation 4 |
> > | ---------------- | ---------- | ---------- | ---------- | ---------- |
> > | Ground Truth ΔΔG | -0.5       | 0.5        | 10         | 15         |
> > | DDGPred          | 0.4        | -0.3       | 11         | 16         |
> > | RDE              | -0.6       | 0.4        | 9          | 7          |
> >
> > In this toy example, the RMSE, MAE, and Pearson of DDGPred are 0.929, 0.925, 0.996; and those of RDE are 4.032, 2.300, 0.901, which are worse than DDGPred. We can see that large values dominate RMSE, MAE, and Pearson.
> >
> > Nevertheless, in reality, most mutations do not have large ΔΔGs, and the real challenge is to discriminate which improves binding and which does not. In addition, for mutations with large ΔΔG, it is less important to know the exact value as long as we know it has a significant impact on binding.
> >
> > In the above example, DDGPred misclassifies Mutation 1 and 2 but still achieves low RMSE, MAE, and high Pearson. In contrast, RDE classifies all the mutations correctly but has worse scores. This shows that RMSE, MAE, and Pearson are less reliable when outliers exist.
> >
> > There exist outliers in our test set. However, if we remove those outliers ($|\Delta\Delta G| \ge 5$), RMSE, MAE, and Pearson scores of DDGPred drop and become worse than those of RDE:
> >
> > | (Outliers Removed) | Overall Pearson | Overall Spearman | RMSE       | MAE        | AUROC      |
> > | ------------------ | --------------- | ---------------- | ---------- | ---------- | ---------- |
> > | **DDGPred**        | 0.4676          | 0.4248           | 1.3111     | 0.9951     | 0.6861     |
> > | **RDE-Network**    | **0.5205**      | **0.4949**       | **1.2665** | **0.9460** | **0.7137** |
> >
> > ----------

---

> > > ### Author Response · Authors · 2022-11-13
> > > **Response (3/3)**
> > >
> > >
> > > **[Q6] Biophysical background of conformational entropy**
> > >
> > > [1] and [2] introduce the relationship between entropy and $\Delta \Delta G$ concisely. We summarize key points below:
> > >
> > > - The physical quantity for measuring the binding strength between two groups of protein is the Gibbs free energy of **association** $\Delta G_a = \Delta H - T\Delta S$.
> > >   - $\Delta H$ is the change in enthalpy upon the formation of the complex. Generally, as no covalent bonds are formed or broken upon protein-protein binding, we consider no change in enthalpy and ignore $\Delta H$.
> > >   - $\Delta S$ is the change in entropy upon binding. It is defined as $\Delta S = S_a - S_d$, where $S_a$ is the entropy of the proteins in the bound state (complex), and $S_d$ is the entropy in the unbound state (separated).
> > > - The entropy $S$ is given by the Boltzmann expression $S = -k_B \int p(\mathbf{x}) \log p(\mathbf{x})\mathrm{d}\mathbf{x} = -k_B \mathbb{E}_{\mathbf{x}\sim p}\log p(\mathbf{x})$. Here $p(\mathbf{x})$ is the distribution of conformation $\mathbf{x}$.
> > > - In this work, we use the Rotamer Density Estimator to approximate $p(\mathbf{x})$. The entropy $S$ can be estimated by the entropy of the RDE, denoted by $H$ (Eq.9), up to a negative constant coefficient $-k_B$.
> > >   - $S \approx -k_B H$. Note that the sign of the Boltzmann entropy is different from the sign of the model entropy.
> > > - Since $\Delta S = S_a - S_d$, $\Delta S$ can be estimated by subtracting the estimation of $S_d$ from the estimation of $S_a$. This leads to Eq.10 of the paper.
> > >   - The entropy of the bound state $S_a$ is estimated by considering the whole protein complex.
> > >   - The entropy of the separated state $S_d$ is given by the sum of the estimated entropy of the two separated groups of proteins.
> > > - The change in binding free energy is given by $\Delta\Delta G = \Delta G^\text{mut} - \Delta G^\text{wt} = - T(\Delta S^\text{mut} - \Delta S^\text{wt}) \propto -(S^\text{mut}_a - S^\text{mut}_d - S^\text{wt}_a + S^\text{wt}_d)$.  We estimate $\Delta \Delta G$ by substituting $S$ with $-k H$, which leads to Eq.12.
> > >
> > > [1] Brady, G. P., & Sharp, K. A. (1997). Entropy in protein folding and in protein-protein interactions. *Current opinion in structural biology*, *7*(2), 215-221.
> > >
> > > [2] Kastritis, P. L., & Bonvin, A. M. (2013). On the binding affinity of macromolecular interactions: daring to ask why proteins interact. *Journal of The Royal Society Interface*, *10*(79), 20120835.
> > >
> > > ----------
> > >
> > > **[Q7] Training data**
> > >
> > > Protein structures determined by crystallography have been used to derive statistics of sidechain conformations for decades [3-5]. In principle, structures determined by Cryo-EMs should be more faithful, but high-resolution Cryo-EM structures are much fewer. As far as we are concerned, there do not seem to be any choices better than crystallography.
> > >
> > > [3] Dunbrack Jr, R. L., & Karplus, M. (1993). Backbone-dependent rotamer library for proteins application to side-chain prediction. *Journal of molecular biology*, *230*(2), 543-574.
> > >
> > > [4] Dunbrack Jr, R. L. (2002). Rotamer libraries in the 21st century. *Current opinion in structural biology*, *12*(4), 431-440.
> > >
> > > [5] Shapovalov, M. V., & Dunbrack Jr, R. L. (2011). A smoothed backbone-dependent rotamer library for proteins derived from adaptive kernel density estimates and regressions. *Structure*, *19*(6), 844-858.

---

### Official Review · Reviewer_GvK9 · 2022-10-24

**Confidence:** 4
**Correctness:** 3
**Technical Novelty And Significance:** 4
**Empirical Novelty And Significance:** 3
**Recommendation:** 8

**Clarity, Quality, Novelty And Reproducibility:**

**Clarity**
- The paper is well written and structured.
- Related work is adequately referenced, and proper background is provided to the reader to understand the work

**Quality**
- Sound experiment design: comprehensive set of performance metrics; limited overlap between folds to mitigate data leakage between train and test sets
- Strong empirical results relative to baselines
- Some claims not fully substantiated (see above)

**Novelty**
- The use of normalizing flow for exact rotamer density estimates is novel

**Reproducibility**
- Code not provided at submission. Authors confirmed in appendix A.4 that all data and code will be made available once the paper is made public


**Strength And Weaknesses:**

**Strengths**
- Very clear write up, with a well-thought-through experimental design
- From a methodology standpoint, the unsupervised modeling approach via RDE avoids potential issues due to label scarcity / bias. Furthermore, the choice of normalizing flow allows efficient sampling of rotamers when estimating the different entropy terms since it enables exact likelihood estimation
- The method achieves very strong performance relative to other baselines, without relying on structure data as an input

**Weaknesses**
- While it seems intuitively sensible, the claim that sequence-based models fail to predict ∆∆G for protein-protein binding does not seem particularly well backed-up experimentally. To achieve good performance with ESM-1v, one must both: a) use an ensemble of 5 ESM1v transformers b) fine-tune each model on a set of homologous sequences. When both conditions are not met (and it seems it is not the case here), a single non fine-tuned ESM-1v performs worse than a site independent model (see Tables 1 and 2 of [1], or table 2 of [2]). More performing baselines include Tranception [2], EVE [3] or MSA Transformer [4].
- Table 1: it seems that DDGPred does best in terms of both RMSE and MAE (bolding issue?)

----------------------------------------------------------------------------------------
[1] Meier, J., Rao, R., Verkuil, R., Liu, J., Sercu, T., & Rives, A. (2021). Language models enable zero-shot prediction of the effects of mutations on protein function. NeurIPS.

[2] Notin, P., Dias, M., Frazer, J., Marchena-Hurtado, J., Gomez, A.N., Marks, D.S., & Gal, Y. (2022). Tranception: Protein Fitness Prediction with Autoregressive Transformers and Inference-time Retrieval. ICML.

[3] Frazer, J., Notin, P., Dias, M., Gomez, A.N., Min, J.K., Brock, K.P., Gal, Y., & Marks, D.S. (2021). Disease variant prediction with deep generative models of evolutionary data. Nature.

[4] Rao, R., Liu, J., Verkuil, R., Meier, J., Canny, J.F., Abbeel, P., Sercu, T., & Rives, A. (2021). MSA Transformer. ICML.


**Summary Of The Paper:**

This paper focuses on quantifying the effects of mutations on protein-protein interactions. It does so by first building a (conditional) normalizing flow model to estimate the probability of sidechain conformations (rotamers) conditioning on the type, position, orientation, and prior rotamer of itself and other amino acids. Building on the intuition that higher binding affinity between two proteins implies higher rigidity (lower entropy), it then estimates the change in binding free energy due to the mutation as the change in delta entropy (bound - unbound) between mutated sequence and wild type. Experiments on the SKEMPI2 dataset illustrate the strengths of the method over current baselines.

**Summary Of The Review:**

A compelling approach to quantify the effects of mutations in PPI, which does not rely on labels for rotamer density estimates and does not need structure input to make predictions. The approach achieves strong empirical performance relative to baselines. I would be willing to recommend acceptance more enthusiastically if the concerns discussed above are addressed during rebuttal.

------------------------------------------------------------------------------------------------------------
[UPDATE POST REBUTTAL]
Authors' responses adequately addressed my feedback during rebuttal. Updating my score accordingly.

---

> ### Author Response · Authors · 2022-11-13
> **Response**
>
> We thank the reviewer’s constructive comment on our work! Below are our responses to the reviewer's concerns and clarification on some issues.
>
> ---
>
> **[Q1] Additional sequence/evolution-based baselines**
>
> As suggested by the reviewer, we have additionally compared our RDE method with three evolution-based methods, including PSSM, MSA-Transformer, and Tranception. Following [1], we built MSAs with Jackhmmer from the Uniref90 database and applied HHfilter before feeding the MSAs to PSSM and MSA-Transformer. Details about the baselines are updated in Section A.3 of the appendix. Results are also updated in Table 1.
>
> According to the results, we find that there is still a large gap between sequence-based methods and structure-based methods on PPI mutation effect prediction. This further evidences our argument in Section 2.2 --- sequence-based methods are not suitable for most PPIs.
>
> After we dig deeper into the results of sequence-based methods, we have the following findings:
>
> - Under circumstances where we have little co-evolutionary information, such as antibody-antigen binding, sequence-based methods are more likely to fail. We select antibody-antigen complexes from the SKEMPI datasets and evaluate per-structure Spearman correlation coefficients of the prediction by MSA Transformer and RDE-Network on these antibody-antigen complexes. The average Spearman score of MSA Transformer is 0.0744, and the average score of RDE-Network is 0.4284.
> - When the proteins in a complex come from the same organism, evolution-based methods are more likely to be effective since proteins in such a complex usually function as a whole in the organism, so they evolve together. We inspect 10 complexes on which MSA Transformer performs the best in terms of Spearman coefficients and find that 9 of them consist of proteins from a single organism.
>
> We put the detailed analysis of the above findings along with plots and tables in **Section B.1 (a new section)** of the appendix. We invite the reviewer to take a look at the section, and we would be glad about further comments and discussions!
>
> [1] Meier, J., Rao, R., Verkuil, R., Liu, J., Sercu, T., & Rives, A. (2021). Language models enable zero-shot prediction of the effects of mutations on protein function. NeurIPS.
>
> ---
>
> **[Q2] About ESM-1v**
>
> We used the official script for variant prediction based on ESM-1v. It already uses an ensemble of 5 different transformers. ESM-1v can be fine-tuned using MSAs. However, we have run three MSA-based methods as baselines, so we believe they can represent the ability of MSAs in PPI mutation effect prediction. In addition, it is unaffordable for us to fine-tune ESM-1v: the SKEMPI dataset contains 347 different structures, which means we would need to train 347 transformers.
>
> ---
>
> **[Q3] Why does DDGPred seem to perform better in terms of RMSE, MAE, and Pearson**
>
> This is because RMSE, MAE, and Pearson are sensitive to outlying ΔΔG values (whose absolute values are too large), so it overestimates the accuracy of DDGPred.
>
> To understand what happens, we give the following illustrative example:
>
> |                  | Mutation 1 | Mutation 2 | Mutation 3 | Mutation 4 |
> | ---------------- | ---------- | ---------- | ---------- | ---------- |
> | Ground Truth ΔΔG | -0.5       | 0.5        | 10         | 15    |
> | DDGPred          | 0.4        | -0.3       | 11         | 16    |
> | RDE              | -0.6       | 0.4        | 9          | 7     |
>
> In this toy example, the RMSE, MAE, and Pearson of DDGPred are 0.929, 0.925, 0.996; and those of RDE are 4.032, 2.300, 0.901, which are worse than DDGPred. We can see that large values dominate RMSE, MAE, and Pearson.
>
> Nevertheless, in reality, most mutations do not have large ΔΔGs, and the real challenge is to discriminate which improves binding and which does not. In addition, for mutations with large ΔΔG, it is less important to know the exact value as long as we know it has a significant impact on binding.
>
> In the above example, DDGPred misclassifies Mutation 1 and 2 but still achieves low RMSE, MAE, and high Pearson. In contrast, RDE classifies all the mutations correctly but has worse scores. This shows that RMSE, MAE, and Pearson are less reliable when outliers exist.
>
> There exist outliers in our test set. However, if we remove those outliers ($|\Delta\Delta G| \ge 5$), RMSE, MAE, and Pearson scores of DDGPred drop and become worse than those of RDE:
>
> | (Outliers Removed) | Overall Pearson | Overall Spearman | RMSE       | MAE        | AUROC      |
> | ------------------ | --------------- | ---------------- | ---------- | ---------- | ---------- |
> | **DDGPred**        | 0.4676          | 0.4248           | 1.3111     | 0.9951     | 0.6861     |
> | **RDE-Network**    | **0.5205**      | **0.4949**       | **1.2665** | **0.9460** | **0.7137** |
>
> ---
>
> **[Q4] Formatting issues**
>
> We have fixed the bolding issues in Table 1 and Table 2. We have also rearranged Table 1, 6, and 7 for better clarity.

---

> > ### Comment · Reviewer_GvK9 · 2022-12-09
> > **Re: responses**
> >
> > Dear authors,
> >
> > Thank you for the thorough responses. I found the response to [Q1] and the corresponding deep dive in B1 to be particularly insightful. No further comments on the other points of feedback. I have raised my score accordingly.

---

### Official Review · Reviewer_R6Cf · 2022-11-01

**Confidence:** 3
**Correctness:** 3
**Technical Novelty And Significance:** 2
**Empirical Novelty And Significance:** 3
**Recommendation:** 6

**Clarity, Quality, Novelty And Reproducibility:**

The paper start with a very clear introduction but then since the method has multiple building blocks it becomes harder to follow.

I have some questions for the authors:

1- Are there any protein overlaps between X-ray structures in PDB used for training the rotamer density estimator and SKEMPI2 database? How can we make sure there is no data leakage?

2- Are the ESM-1v and other baselines fine tuned on the same training data used in RDE?

3- Can the authors elaborate on the two mutations in Table 2 that do not rank above top-10%? Why does the algorithm fail in those cases?

4- Is there a regime where DDGPred is more favorable compared to RDE?

5- Results in Table. 3 show statistical significance however the correlation coefficients are still not high between RDE and entropy. Is rank correlation a better measure?


**Strength And Weaknesses:**

Strength.
1- The paper tackles an important problem. Predicting protein-protein interactions in a low-data regime is a very important problem.

2- No direct homology and evolutionary information is required in the model and thus the method is applicable to find the effect of mutations on protein-protein interactions with a lack of homology information.

3- Empirical results in Table 1 are convincing (although I have some questions about the baselines).

4- The flow-based generative model works on exact likelihood.

5- Using side-chain conformational change as a proxy for delta delta G seems to be novel.

Weakness.
1- The approach only takes into account flexibility in the side chains and not the backbone.

2- If mutations cause change beyond side-chain conformational variations it is not clear how the method can handle those mutations.

3- The significance of the contribution to the ML community is limited. The work tackles a specific problem in protein-protein interactions and mostly uses a combination of standard DL modules (flow based generative models, MLP, attention with rotation/translation invariance) as building blocks.


**Summary Of The Paper:**

The paper develops a flow-based generative approach to model the probability distribution of side-chain conformations (called RDE) in protein-protein interactions and then estimates changes in entropy by measuring the changes in side-chain flexibility. The paper uses a conditional generative model built upon normalizing flows for estimating densities. It uses a neural network to extract delta delta G from the RDE representation.

**Summary Of The Review:**

Overall the paper tackles an important problem and some of the empirical results are very string however in terms of methodology it is not a significant contribution to machine learning.

---

> ### Author Response · Authors · 2022-11-13
> **Response (1/2)**
>
>
> We appreciate the reviewer’s valuable comment on our work! Below are our responses to the reviewer's concerns and clarification on some issues.
>
> ---
>
> **[Q] About data overlap**
>
> Training the Rotamer Density Estimator only requires protein structures, which are **unlabeled data** that does not contain information about binding affinity. There is an overlap between structures in SKEMPI and structures in the training split of PDB-REDO but this cannot be regarded as data leakage.
>
> In other words, let's denote a set of protein structures by $\{X_i\}$ and their binding affinity by $\{y_i\}$. As our goal is to predict $y$ from $X$, we cannot say there is data leakage if we pre-train our network using only $\{X_i\}$. Take ESM-1v, for example. It is trained on Uniref90 and tested on several DMS datasets. The sequences in the DMS datasets are contained in Uniref90, but as ESM-1v is not trained using protein fitness values, we cannot say there is data leakage.
>
> At the finetuning stage, since we leverage ground truth values in SKEMPI, we split the SKEMPI dataset *by structure* to ensure no data overlap (described in Section 3.5). To evaluate the RDE's ability to predict sidechain conformations, we also construct a test split from PDB-REDO that does not share similar sequences with the training split (sequence identity cutoff=50%, described in Section 3.5).
>
> ----------
>
> **[Q] Are baselines finetuned?**
>
> Yes, they are fine-tuned using the same data as RDE.
>
> ----------
>
> **[Q] Why does the algorithm fail to identify NH57L and RH103M?**
>
> The RDE's average estimated ΔΔG of NH57L is -0.16. It is slightly less than 0, and RDE classifies it as stabilizing mutation correctly though not confident.
>
> The RH103M mutation replaces a charged amino acid, arginine (R), with an uncharged hydrophobic amino acid, methionine (M). Eliminating a charged amino acid might enhance the stability of the antibody. Adding a hydrophobic amino acid on the surface is generally considered favorable to neutralizing antigens due to the hydrophobicity effect. These might explain why the mutation improves the neutralization. However, these factors do not directly account for the change in binding free energy. Therefore, our model, as well as other baselines prone to fail.
>
> ------
>
> **[Q] Why does DDGPred seem to perform better in terms of RMSE, MAE, and Pearson**
>
> This is because RMSE, MAE, and Pearson are sensitive to outlying ΔΔG values (whose absolute values are too large), so it overestimates the accuracy of DDGPred.
>
> To understand what happens, we give the following illustrative example:
>
> |                  | Mutation 1 | Mutation 2 | Mutation 3 | Mutation 4 |
> | ---------------- | ---------- | ---------- | ---------- | ---------- |
> | Ground Truth ΔΔG | -0.5       | 0.5        | 10         | 15         |
> | DDGPred          | 0.4        | -0.3       | 11         | 16         |
> | RDE              | -0.6       | 0.4        | 9          | 7          |
>
> In this toy example, the RMSE, MAE, and Pearson of DDGPred are 0.929, 0.925, 0.996; and those of RDE are 4.032, 2.300, 0.901, which are worse than DDGPred. We can see that large values dominate RMSE, MAE, and Pearson.
>
> Nevertheless, in reality, most mutations do not have large ΔΔGs, and the real challenge is to discriminate which improves binding and which does not. In addition, for mutations with large ΔΔG, it is less important to know the exact value as long as we know it has a significant impact on binding.
>
> In the above example, DDGPred misclassifies Mutation 1 and 2 but still achieves low RMSE, MAE, and high Pearson. In contrast, RDE classifies all the mutations correctly but has worse scores. This shows that RMSE, MAE, and Pearson are less reliable when outliers exist.
>
> There exist outliers in our test set. However, if we remove those outliers ($|\Delta\Delta G| \ge 5$), RMSE, MAE, and Pearson scores of DDGPred drop and become worse than those of RDE:
>
> | (Outliers Removed) | Overall Pearson | Overall Spearman | RMSE       | MAE        | AUROC      |
> | ------------------ | --------------- | ---------------- | ---------- | ---------- | ---------- |
> | **DDGPred**        | 0.4676          | 0.4248           | 1.3111     | 0.9951     | 0.6861     |
> | **RDE-Network**    | **0.5205**      | **0.4949**       | **1.2665** | **0.9460** | **0.7137** |

---

> > ### Author Response · Authors · 2022-11-13
> > **Response (2/2)**
> >
> > **[Q] About the regression coefficients**
> >
> > The magnitude of regression coefficients depends on the unit we choose. The unit of ΔΔG in SKEMPI is kcal/mol. If we change the unit to kJ/mol (1kcal/mol = 4.184kJ/mol), the table becomes:
> >
> > | Var.                    | Sign | Coef.  | P-value         | Signif. |
> > | ----------------------- | ---- | ------ | --------------- | ------- |
> > | $H_{M L}^\text{bound}$ | +    | 2.1430 | < 0.001 | ***     |
> > | $H_{M R}^\text{bound}$ | +    | 0.7565 | 0.005           | **      |
> > | $H_{M L}^\text{unbnd}$ | -    | 1.1920 | < 0.001 | ***     |
> > | $H_{W L}^\text{bound}$ | -    | 2.0665 | < 0.001 | ***     |
> > | $H_{W R}^\text{bound}$ | -    | 1.0523 | < 0.001 | ***     |
> > | $H_{W L}^\text{unbnd}$ | +    | 1.0339 | < 0.001 | ***     |
> > | $H_{R}^\text{unbnd}$    | /    | 0.1360 | 0.230           | -       |
> > | Bias                    | /    | 0.7899 | < 0.001 | ***     |
> >
> > We can make the coefficients even higher by changing the unit to J/mol. Therefore, we cannot test the relationship between the entropies and ΔΔG according to the value of coefficients that are unit-dependent. Statistical significance is more suitable for us. The purpose of coefficients in our case is to see whether the sign (direction of contribution) matches the thermodynamic definition of ΔΔG, and the conclusion is that they match.
> >
> > ----------
> >
> > **[Q] About backbone flexibility**
> >
> > One of the limitations of RDE and other ΔΔG predictors is that they cannot model backbone flexibility. However, we believe we can extend RDE to model backbones. The backbone conformation of an amino acid can be parameterized by two angles: $\phi, \psi \in [-\pi, \pi]$. Therefore, we can augment the RDE to model these two additional angles to allow backbone flexibility. Admittedly, there must be other challenges that we should consider when modeling backbones, so we think it deserves effort in future work.

---

### Decision · Program_Chairs · 2023-01-20

**Decision:**

Accept: poster

**Justification For Why Not Higher Score:**

Two of the reviewers still had some minor issues so a poster is fine. The topic is although definitely relevant for ICLR also somewhat speciailzed.

**Justification For Why Not Lower Score:**

Could be lower but still the paper is ready to ship so to speak.

**Metareview: Summary, Strengths And Weaknesses:**

The paper is about density modeling of rotamers (biomolecule degrees of freedom).

All reviewers appreciated the paper and the authors and reviewers engaged in a good discussion further supporting that this is worthwhile work,

**Note From Pc:**

if the above contains the word "oral" or "spotlight" please see: "oral" presentation means -> notable-top-5% and "spotlight" means -> notable-top-25%. As stated in our emails, we are disassociating presentation type from AC recommendations